# ABConformer: Physics-inspired Sliding Attention for Antibody-Antigen Interface Prediction

## Abstract

Accurate prediction of antibody-antigen (Ab-Ag) interfaces is critical for vaccine design, immunodiagnostics and therapeutic antibody development. However, achieving reliable predictions from sequences alone remains a challenge. In this paper, we present ABConformer, a model based on the Conformer backbone that captures both local and global features of a biosequence. To accurately capture Ab-Ag interactions, we introduced the physics-inspired sliding attention, enabling residue-level contact recovery without relying on three-dimensional structural data. ABConformer can accurately predict paratopes and epitopes given the antibody and antigen sequence, and predict pan-epitopes on the antigen without antibody information. In comparison experiments, ABConformer achieves state-of-the-art performance on a recent SARS-CoV-2 Ab-Ag dataset, and surpasses widely used sequence-based methods for antibody-agnostic epitope prediction. Ablation studies further quantify the contribution of each component, demonstrating that, compared to conventional cross-attention, sliding attention significantly enhances the precision of epitope prediction. To facilitate reproducibility, we will release the code under an open-source license upon acceptance.

## 1 Introduction

Antibodies are Y-shaped glycoproteins with two arms (Fab fragments) and a stem (Fc fragment), where the arms contain antigen-binding sites at their tips and are connected to the stem through a flexible hinge. Each antibody has two identical heavy (Ab-H) and light (Ab-L) chains, with each chain containing an N-terminal variable domain (VH in heavy, VL in light) followed by constant domains (CH1–CH3 in heavy, CL in light). Within the variable domains, three hypervariable loops from the heavy chain and three from the light chain—called complementarity-determining regions (CDRs)—cluster together at the tip of the Fab to form the antigen-binding site, a spatially contiguous surface on the antibody (paratopes) that engages the corresponding binding sites on the antigen (epitopes), together forming the antibody-antigen (Ab-Ag) interfaces.

Identifying Ab-Ag interfaces is critical for vaccine design (Tarrahimofrad et al., 2021; Sarvmeili et al., 2024), disease diagnosis (Ricci et al., 2023; Bourgonje et al., 2023), antibody engineering (Kumar et al., 2024; Fantin et al., 2025) and research into immune evasion (Nabel et al., 2021; Liu et al., 2022; Dejnirattisai et al., 2022; Liu et al., 2024), autoimmunity (Curran et al., 2023; Michalski et al., 2024; Iversen et al., 2025) and immunotherapy (Bonaventura et al., 2022; Casirati et al., 2023; Shah et al., 2025). Experimental techniques such as X-ray crystallography and cryo-electron microscopy provide high-resolution Ab-Ag interactions but are resource-intensive (Brändén & Neutze, 2021; Rubach et al., 2025). Phage display is faster but lacks atomic-level precision (Ledsgaard et al., 2022). Therefore, many *in silico* methods have been developed to predict Ab-Ag interfaces.

Current computational methods for predicting Ab-Ag interfaces mainly follow two directions. The first focuses on predicting interfaces using information from both antibodies and antigens. Representative methods, including PECAN (Pittala & Bailey-Kellogg, 2020), Honda (Honda et al., 2020), Epi-EPMP (Del Vecchio et al., 2021), PeSTo (Krapp et al., 2023), SEPPA-mAb (Qiu et al., 2023), MIPE (Wang et al., 2024), DeepInterAware (Xia et al., 2025) and Epi4Ab (Tran et al., 2025), have shown strong performance in predicting antibody-specific interfaces. The second direction aims

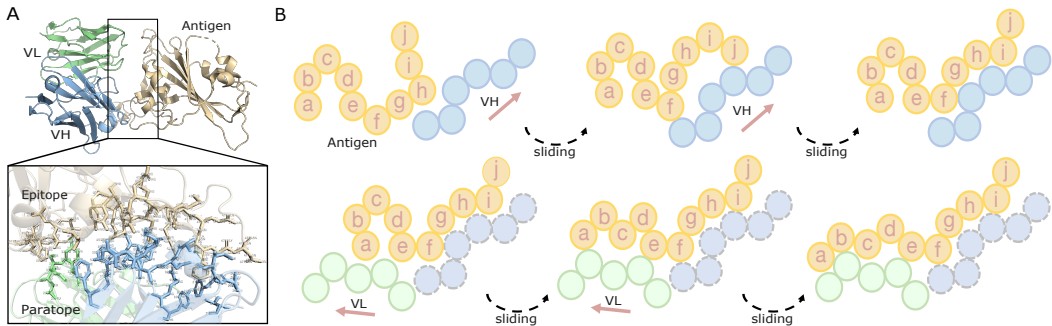

Figure 1: Interfaces sliding process. (A) Visualization of the SARS-CoV-2 Omicron BA.1 RBD in complex with the CAB-A17 antibody (PDB ID: 8C0Y). Interfaces are identified using a 4 Å distance cutoff (Appendix A). (B) Schematic of the interfaces sliding process.

to predict pan-epitopes on antigens in the absence of antibody information, thereby facilitating *de novo* antibody design for new antigens. Widely-adopted approaches, such as BepiPred-3.0 (Clifford et al., 2022), DiscoTope-3.0 (Høie et al., 2024) and SEMA 2.0 (Ivanisenko et al., 2024), have achieved comparatively better performance in large-scale B-cell epitope prediction.

However, accurate prediction of Ab-Ag interfaces remains challenging for several reasons. First, except Epi4Ab, current antibody-specific methods treat the antibody input as a whole without distinguishing heavy and light chains, which lacks physical interpretability as paratopes are formed by hypervariable loops from both VH and VL domains (Fig. 1A). Second, although some models (*e.g.*, Honda) employ cross-attention to capture Ab-Ag interactions, they struggle with dependencies that may be distracted by distant, irrelevant positions, given that Ab–Ag interfaces are confined to specific regions rather than spanning the entire sequence. Third, antibody-agnostic epitope prediction are limited by the scarcity of experimentally solved 3D structures. Although BepiPred-3.0 and SEMA-1D 2.0 are sequence-based methods, they underperform compared to structure-based or multi-modal methods.

Therefore, we design a sequence-based method that represents the Ab–Ag complex as three components—Ab-H, Ab-L and Ag—to predict Ab–Ag interfaces when antibodies are provided, and pan-epitopes from antigen alone. To capture both local patterns and long-range dependencies of a single biosequence, we adopt the Conformer architecture that combines convolution and self-attention (Gulati et al., 2020). To further capture interactions between biosequences, we introduce sliding attention into our model (Feng et al., 2024). Unlike conventional cross-attention, sliding attention accounts for spatial proximity and iteratively adjusts relative positions between two sequences, thereby uncovering more stable interaction patterns. In our cases, the antigen sequence first slides against Ab-H, and then Ab-L, generating an attention map for each sliding process (Fig. 1B).

To summarize, we propose **ABConformer**, an interfaced-based explainable AntiBody target prediction model with physics-inspired sliding-attention Conformer architecture. ABConformer has several advantages. First, it achieves a comprehensive improvement in predicting antibody-specific interfaces, while also outperforming all sequence-based methods in identifying antibody-agnostic epitopes on the SARS-CoV-2 dataset filtered from 2024 onwards. Second, it simulates the molecular docking process, providing a physically interpretable view of Ab–Ag interactions and pairwise residue relationships. Third, it enables large-scale prediction of Ab–Ag interfaces in the absence of 3D structures, which is particularly valuable in vaccine development, where numerous viral variants, multiple antigenic targets and candidate antibodies need to be assessed.

## 2 METHODS

### 2.1 SLIDING ATTENTION

Sliding attention is motivated by the physical process of molecular docking, where a biosequence dynamically slides along its partner to maximize the stability of interactions (Feng et al., 2024). It computes attention from both feature similarity and spatial proximity, iteratively updating antigen

residues first along the interaction gradients of Ab-H, then along those of Ab-L, thereby accurately capturing the features of Ab–Ag interfaces. An algorithm is provided in Appendix B.

**Feature attention.** Consider a sliding sequence $X^{(t)} = \{x_1^{(t)}, x_2^{(t)}, \ldots, x_m^{(t)}\}$ and a reference sequence $Y^{(t)} = \{y_1^{(t)}, y_2^{(t)}, \ldots, y_n^{(t)}\}$, where $t$ is the iteration step and the residue embeddings satisfy $x_i^{(t)}, y_j^{(t)} \in \mathbb{R}^d$. To capture the feature similarity, embeddings are first projected into learnable latent spaces using linear maps $E_S, E_R \in \mathbb{R}^{d \times d}$, which yields the projected embeddings $X^{(t)} E_S \in \mathbb{R}^{m \times d}$ and $Y^{(t)} E_R \in \mathbb{R}^{n \times d}$. The pairwise attention score $A_{ij}^{(t)}$ is then computed as:

$$a_{ij}^{(t)} = \frac{(x_i^{(t)} E_S) \cdot (y_j^{(t)} E_R)^\top}{\sqrt{d}},$$
$$A_{ij}^{(t)} = \exp\left( a_{ij}^{(t)} - \max_{k \in [1,n]} a_{ik}^{(t)} \right). \tag{1}$$

Here, each row of the scaled dot-product matrix is shifted by its maximum to prevent numerical overflow. The exponential scores then lie in $(0, 1]$, providing non-negative affinities between residues.

**Spatial attention.** The spatial proximity matrix $S^{(t)} \in \mathbb{R}^{m \times n}$ is estimated using a Gaussian kernel over the sequence positions. Assuming that the reference positions $Q = (q_1, \ldots, q_n)$ are fixed integers along $Y$, and the sliding positions $P^{(t)} = (p_1^{(t)}, \ldots, p_m^{(t)})$ are learnable positions of $X$ at iteration $t$, the spatial attention score $S_{ij}^{(t)}$ is written as:

$$S_{ij}^{(t)} = \exp\left( -\frac{(p_i^{(t)} - q_j)^2}{2h^2} \right). \tag{2}$$

Here, $h$ is the bandwidth determined by the length of the reference sequence $Y$. A smaller $h$ restricts the receptive field, causing sliding residues at $p_i^{(t)}$ to be attracted to less distant residues in $Y$, thereby confining each sliding process to a specific region. Assuming a binary mask $M \in \{0, 1\}^{m \times n}$, where $M_{ij} = 1$ if $(i, j)$ is valid and $0$ if padding. The bandwidth $h$ is determined by the valid length of $Y$, scaled by a factor $c$, and constrained to the range $[h_{\min}, h_{\max}]$:

$$h = \min\left\{ h_{\max}, \ \max\left\{ h_{\min}, \ \sum_{j=1}^{n} \frac{M_{:,j}}{c} \right\} \right\}. \tag{3}$$

**Weighted attention.** After obtaining feature and spatial attention, the weighted attention matrix is computed as the Hadamard product of them:

$$W_{ij}^{(t)} = M_{ij} (A^{(t)} \odot S^{(t)})_{ij}. \tag{4}$$

Here, $W$ captures the combined affinity between residues of the sliding and reference sequences, with higher values indicating stronger potential interactions. Since $W$ is unnormalized, we perform row-wise and column-wise normalization to convert it into convex combination weights suitable for attention aggregation:

$$\widehat{W}_{ij}^{(t)} = \frac{W_{ij}^{(t)}}{\sum_{k=1}^{n} W_{ik}^{(t)} + \varepsilon}, \qquad \widetilde{W}_{ij}^{(t)} = \frac{W_{ij}^{(t)}}{\sum_{k=1}^{m} W_{kj}^{(t)} + \varepsilon}, \tag{5}$$

where $\varepsilon$ is a small constant added for numerical stability. Row-normalization ensures that each sliding residue $x_i$ distributes its attention over the reference residues $y_j$, and column-normalization guarantees that each reference residue aggregates contributions from all sliding residues.

**Embedding updates.** Using the normalized attention weights, residue embeddings are iteratively updated via cross-attention with residual connections:

$$X^{(t+1)} = \widehat{W}^{(t)} (Y^{(t)} E_Y) + X^{(t)},$$
$$Y^{(t+1)} = (\widetilde{W}^{(t)})^\top (X^{(t)} E_X) + Y^{(t)}. \tag{6}$$

Here, $E_X, E_Y \in \mathbb{R}^{d \times d}$ are linear projections mapping embeddings into value spaces. Each sliding residue in $X^{(t)}$ queries all residues in $Y^{(t)}$ through $\widehat{W}^{(t)}$, aggregating contextual information, and similarly, each residue in $Y^{(t)}$ aggregates information from $X^{(t)}$ via $\widetilde{W}^{(t)}$.

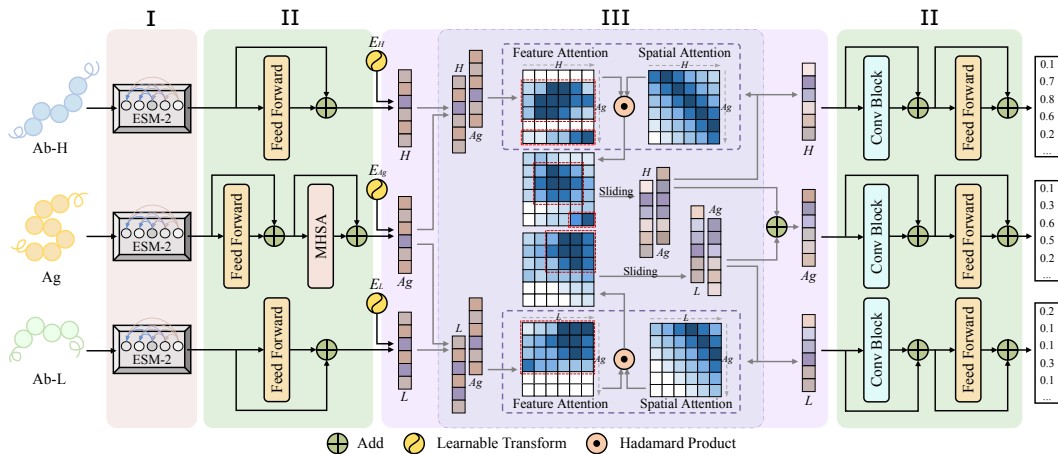

Figure 2: Overview of the ABConformer architecture, comprising (I) an encoding layer, (II) Conformer layers, and (III) sliding attention modules. Six layers of stage II and III are stacked in the standard ABConformer (Appendix E). MHSA denotes multi-head self-attention.

**Position updates.** Finally, the sliding positions themselves are refined according to the attention distribution, which is computed as:

$$P^{(t+1)} = \widehat{W}^{(t)} Q. \tag{7}$$

An equivalent expression of this process is (Appendix B):

$$p_i^{(t+1)} - p_i^{(t)} = \sum_{j=1}^{n} \widehat{W}_{ij}^{(t)} (q_j - p_i^{(t)}). \tag{8}$$

Here, the update can be intuitively understood as each residue in the sliding sequence being 'pulled' toward regions where the reference residues collectively exert stronger interactions. Each reference residue contributes to this movement proportionally to its weighted attention, so residues naturally migrate toward positions of higher cumulative affinity. Conceptually, this process is analogous to mean-shift mode seeking (Cheng, 1995), where each iteration shifts residue $x_i$ along the gradient of an underlying density function. In our case, this density is the accumulated interaction magnitude at the current position: $f(p_i^{(t)}) = \sum_{j}^{n} M_{ij} A_{ij}^{(t)} S_{ij}^{(t)}$. And $x_i$ moves along the gradient of $f(p_i^{(t)})$.

## 2.2 ABCONFORMER

ABConformer adopts a three-branch architecture for Ab-H, Ab-L and the antigen (Fig. 2). In the antigen branch, sequence embeddings are first encoded using ESM-2 150M (Lin et al., 2023), followed by a feedforward layer and a multi-head self-attention (MHSA) module (Vaswani et al., 2017) both with residual connections (He et al., 2016). The antigen embeddings then interact with Ab-H and Ab-L through sliding, iteratively updating both the embeddings (Eq. 6) and sequence positions (Eq. 7). After $T$ steps, this process produces two sets of antigen embeddings, $X_{\mathrm{H}}^{(T)}$ and $X_{\mathrm{L}}^{(T)}$, which are linearly combined as

$$X_{\mathrm{Ag}} = \alpha X_{\mathrm{H}}^{(T)} + (1-\alpha) X_{\mathrm{L}}^{(T)}, \tag{9}$$

where $\alpha \in [0, 1]$ is a weight controlling the contributions of Ab-H and Ab-L. The combined embedding $X_{\mathrm{Ag}}$ is then passed to the remaining Conformer stage, followed by a convolution block and an additional feed-forward layer both with residual connections.

The Ab-H and Ab-L branches are structurally similar, except that the MHSA module is omitted, as it contributes little to paratope prediction when sliding is applied (Appendix G). In the standard ABConformer, six layers of this three-branch backbone (except the encoding part) are stacked, balancing the computational cost with predictive performance (Appendix E).

## 3 EXPERIMENTS

### 3.1 EXPERIMENTS SETUP

**Dataset.** The training set of ABConformer was obtained from AACDB (Zhou et al., 2025), which contains 7,488 experimentally solved structures. A single PDB entry may contain multiple identical complexes arising from repeated copies in the crystal or multiple asymmetric units in the unit cell. To remove redundant entries while retaining a diverse collection of Ab-Ag samples, we selected only one complex per PDB ID, resulting in the final dataset of 3,674 entries. Then we analyzed all antigens from the 3,674 entries and constructed a phylogenetic tree with ClustalOmega (Sievers & Higgins, 2018), resulting in six clusters (Appendix D). Each cluster was then evenly divided into five parts, and one part of each cluster was combined to form a fold. In this way, five folds were generated for cross-validation.

To further evaluate our model compared with other baselines, we extracted an external dataset of SARS-CoV-2 from CoV-AbDab (Raybould et al., 2021). The SARS-CoV-2 set, filtered since 2024, comprises 35 solved structures that has no overlap with the original training data (Appendix D).

**Embedding and Interface Labeling.** Each complex was rigorously decomposed into one Ab-H, Ab-L and Ag chain. Each chain was then embedded using ESM-2 150M to generate a representation of 640 dimensions. Paratopes and epitopes were identified using a 4 Å distance cutoff between heavy atoms of antibody and antigen chains (Van Regenmortel, 2009).

**Training and Evaluation.** ABConformer was initially trained and evaluated via five-fold cross-validation on the AACDB dataset, then retrained on the full dataset to capture more patterns. After retraining, its performance was compared with multiple state-of-the-art methods on the SARS-CoV-2 dataset.

**Performance metrics.** To assess the performance of paratope and epitope predictions, we computed two types of metrics (Appendix C). First, binary classification metrics, including intersection over union (IoU), precision (Prec), recall (Rec), F1 score, and Matthews correlation coefficient (MCC). Second, score-based metrics, including Pearson correlation coefficient (PCC), and the areas under the receiver operating characteristic (ROC) and precision-recall (PR) curves. Higher values of these metrics indicate better predictive performance.

| Target | Method | IoU ↑ | Prec ↑ | Rec ↑ | F1 ↑ | MCC ↑ | PCC ↑ | ROC ↑ | PR ↑ |
|---|---|---|---|---|---|---|---|---|---|
| Ab-Ag *Para* | PECAN | 0.373 | 0.520 | 0.569 | 0.543 | 0.497 | 0.516 | 0.869 | 0.527 |
| | Honda | 0.414 | 0.595 | 0.578 | 0.586 | 0.565 | 0.591 | 0.885 | 0.595 |
| | Epi-EPMP | 0.406 | 0.608 | 0.551 | 0.578 | 0.550 | 0.573 | 0.893 | 0.584 |
| | PeSTo | 0.419 | 0.573 | 0.610 | 0.591 | 0.572 | 0.594 | 0.904 | 0.602 |
| | MIPE | 0.466 | **0.705** | 0.580 | 0.636 | 0.603 | 0.620 | **0.912** | 0.638 |
| | DeepInterAware | 0.430 | 0.645 | 0.563 | 0.601 | 0.585 | 0.605 | 0.907 | 0.614 |
| | Epi4Ab | - | - | - | - | - | - | - | - |
| | AF2 Multimer | 0.403 | 0.527 | **0.630** | 0.574 | 0.542 | - | - | - |
| | ABConformer | **0.482** | 0.693 | 0.613 | **0.651** | **0.622** | **0.632** | 0.904 | **0.651** |
| Ab-Ag *Epi* | PECAN | 0.230 | 0.311 | 0.470 | 0.374 | 0.342 | 0.397 | 0.885 | 0.302 |
| | Honda | 0.260 | 0.340 | 0.517 | 0.413 | 0.407 | 0.458 | 0.914 | 0.357 |
| | Epi-EPMP | 0.248 | 0.329 | 0.505 | 0.398 | 0.389 | 0.441 | 0.897 | 0.341 |
| | PeSTo | 0.243 | 0.307 | 0.539 | 0.391 | 0.379 | 0.424 | 0.907 | 0.326 |
| | MIPE | 0.311 | 0.412 | **0.560** | 0.475 | 0.463 | 0.496 | 0.923 | 0.419 |
| | DeepInterAware | 0.273 | 0.364 | 0.523 | 0.429 | 0.414 | 0.469 | 0.915 | 0.369 |
| | Epi4Ab | 0.305 | 0.423 | 0.521 | 0.467 | 0.457 | 0.493 | 0.928 | 0.415 |
| | AF2 Multimer | 0.215 | 0.275 | 0.496 | 0.354 | 0.307 | - | - | - |
| | ABConformer | **0.336** | **0.467** | 0.545 | **0.503** | **0.492** | **0.510** | **0.931** | **0.441** |
| Ag *Epi* | BepiPred-3.0 | 0.077 | 0.087 | 0.403 | 0.143 | 0.162 | 0.187 | 0.862 | 0.094 |
| | SEMA-1D 2.0 | 0.082 | 0.089 | **0.510** | 0.152 | 0.164 | 0.195 | 0.804 | 0.107 |
| | DiscoTope-3.0 | **0.161** | 0.194 | 0.487 | **0.277** | **0.273** | **0.325** | **0.870** | **0.231** |
| | ABConformer | 0.144 | **0.197** | 0.348 | 0.252 | 0.248 | 0.283 | 0.855 | 0.192 |

Table 1: Comparison of antibody-specific methods (Ab–Ag, evaluated on paratopes and epitopes) and antibody-agnostic methods (Ag, evaluated on epitopes) on the SARS-CoV-2 dataset. The best-performing values are highlighted in bold, and the second-best values are underlined.

## 3.2 COMPARISON EXPERIMENTS

To evaluate the performance of predicting antibody-specific interfaces, we selected PECAN, Honda, Epi-EPMP, PeSTo, MIPE, DeepInterAware and Epi4Ab as baseline methods. Each method was evaluated on the SARS-CoV-2 dataset to assess the performance in predicting Ab-Ag interfaces. Furthermore, since AlphaFold2 Multimer (Mirdita et al., 2022) is widely used for predicting protein complex structures, many previous studies have extracted interfaces based on its structural predictions (DeRoo et al., 2024; Clifford et al., 2025). Here, we also used AlphaFold2 Multimer v3 to model all complexes and identified interface residues with a 4 Å distance cutoff, enabling a direct comparison of ABConformer with commonly used tools.

To further assess pan-epitope prediction on antigens, we compared ABConformer with BepiPred-3.0, DiscoTope-3.0 and SEMA-1D 2.0. Both BepiPred-3.0 and SEMA-1D 2.0 are sequence-based methods for conformational epitope prediction, while DiscoTope-3.0 relies on antigen PDB structures. Here, the input for ABConformer only contains antigen sequences, with antibody embeddings set to zero, yielding a classic Conformer architecture (*i.e.*, the sliding-attention module has no effect) for epitope prediction.

Results show that ABConformer comprehensively improves the prediction of paratopes and epitopes compared to all antibody-specific methods, as measured by IoU, F1, MCC, PCC and PR (Tab. 1). Notably, epitope precision is increased by 0.044 relative to the second-best method, indicating that the sliding process enhances the accurate docking between antigen and antibody chains. Furthermore, when antibody information is ignored, ABConformer outperforms current sequence-based antibody-agnostic methods in pan-epitope prediction across IoU, F1, MCC, PCC and PR (Tab. 1). However, the recall is substantially lower than that of other methods. This is attributed to two factors. First, different methods were trained and evaluated using different datasets and epitope identification protocols (Appendix F). Second, ABConformer trades off recall to achieve a substantial improvement in precision.

## 3.3 ABLATION STUDIES

To dissect the components of ABConformer, we performed ablation studies from three perspectives: encoding, sliding attention mechanism, and Conformer modules, which also correspond to three stages (I, III, II) as shown in Figure 2. We first replaced the ESM-2 encoding with one-hot encoding that represents each residue along with its 15 upstream and downstream neighbors, resulting in a 651-dimensional feature vector (21 dimensions per residue × 31 residues in context window). This dimensionality was slightly higher than the 640-dimensional embeddings produced by ESM-2 150M. Then we compared sliding attention with conventional cross-attention, which lacks distance constraints (Eq. 2) and position updates (Eq. 7), as well as with MHSA without chain interactions. Finally, we ablated the Conformer backbone by selectively removing either the convolutional blocks or the MHSA modules. Each variant was evaluated on the AACDB dataset using cross-validation, with metrics demonstrating the mean values of five folds.

| I: Encoding | | III: Attention | | | II: Conformer | | Ag | | | | | Ab-H | Ab-L |
|---|---|---|---|---|---|---|---|---|---|---|---|---|---|
| one-hot | ESM-2 | self | cross | slide | Conv | MHSA | Prec↑ | Rec↑ | PCC↑ | ROC↑ | PR↑ | PCC↑ | PCC↑ |
| ✗ | ✓ | ✗ | ✗ | ✓ | ✓ | ✓ | **0.660** | 0.546 | **0.611** | **0.906** | **0.589** | **0.741** | **0.697** |
| ✓ | ✗ | ✗ | ✗ | ✓ | ✓ | ✓ | 0.499 | 0.490 | 0.536 | 0.892 | 0.502 | 0.737 | 0.691 |
| ✗ | ✓ | ✓ | ✗ | ✗ | ✓ | ✓ | 0.469 | 0.453 | 0.485 | 0.877 | 0.415 | 0.736 | 0.675 |
| ✗ | ✓ | ✗ | ✓ | ✗ | ✓ | ✓ | 0.543 | **0.588** | 0.581 | 0.903 | 0.562 | 0.739 | 0.691 |
| ✗ | ✓ | ✗ | ✗ | ✓ | ✗ | ✓ | 0.557 | 0.539 | 0.572 | 0.901 | 0.539 | 0.735 | 0.687 |
| ✗ | ✓ | ✗ | ✗ | ✓ | ✓ | ✗ | 0.610 | 0.559 | 0.597 | 0.905 | 0.576 | 0.739 | 0.693 |
| ✗ | ✓ | ✗ | ✗ | ✗ | ✓ | ✗ | 0.460 | 0.447 | 0.484 | 0.859 | 0.411 | 0.732 | 0.667 |

Table 2: Ablation studies of ABConformer on antibody-specific interface prediction. The mean metrics of five-fold cross-validation were evaluated on the AACDB dataset (N=3,674) across different encoding strategies (stage I), attention mechanisms (stage III) and Conformer modules (stage II) (Appendix G).

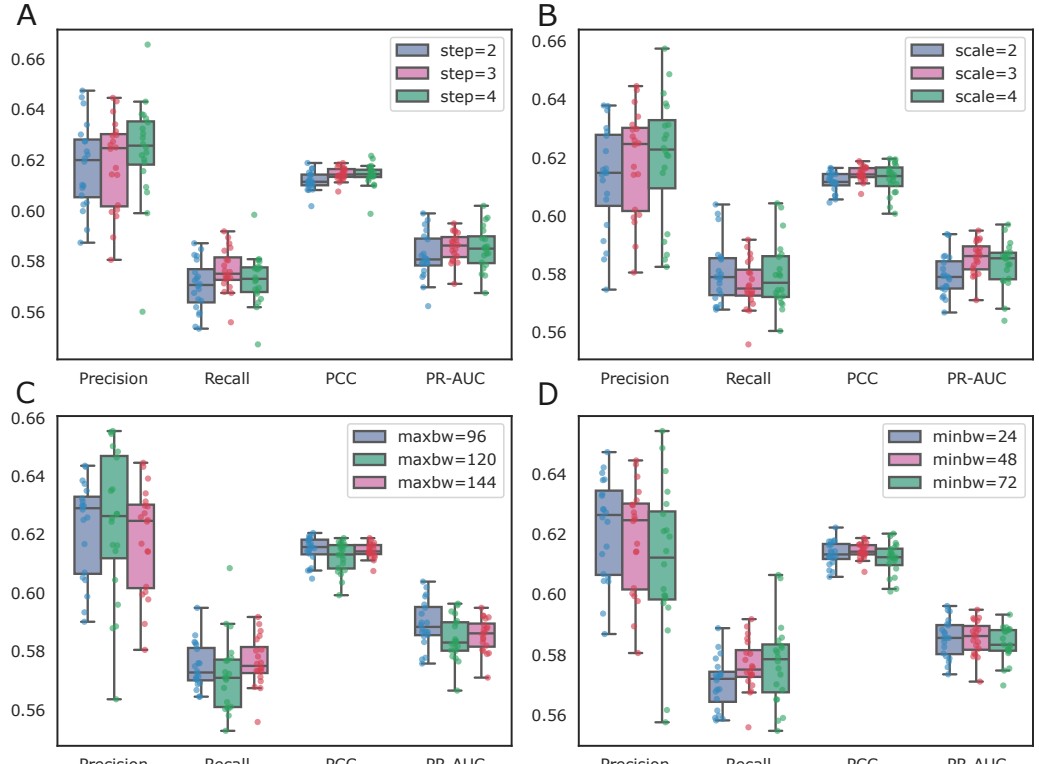

Figure 3: Sensitivity analysis. The box plots show the distribution of metrics from the last twenty training epochs, evaluated on epitopes using the same validation set and random seed (Appendix H). Analyses correspond to (A) sliding step $T$, (B) bandwidth scaling factor $c$, (C) maximum bandwidth $h_{max}$ and (D) minimum bandwidth $h_{min}$ (see METHODS). Default values are indicated in pink.

Here, we analyze the results in two parts: paratope prediction and epitope prediction. For antibody-specific paratope prediction, each variant attains slightly lower performance of paratope prediction on the AACDB dataset (Tab. 2). Variants that remove all attention mechanisms or replace sliding attention with self-attention show notable decreases in predictive performance on Ab-L.

For epitope prediction, each component of ABConformer makes a substantial contribution to the overall performance (Tab. 2). In stage I, ESM-2 embeddings considerably outperform one-hot encoding in predictive performance and input dimensionality. In stage II, removing either convolution blocks or MHSA modules results in modest performance degradation. In stage III, replacing sliding attention with MHSA markedly reduces predictive performance, while substituting it with cross-attention increases recall by 0.042. This is because sliding attention guides antigen residues toward more stable binding configurations limited by the bandwidth, resulting in more conservative scores when two residues are too far apart; while cross-attention distributes interactions across entire sequences, where distant and irrelevant features can inflate attention scores for residues. However, in general, sliding attention achieves superior precision and also outperforms in PCC, ROC and PR.

## 3.4 SENSITIVITY ANALYSIS

In METHODS, we introduced sliding attention along with several hyperparameters, including the number of sliding steps ($T$), the bandwidth scaling factor ($c$), and the maximum and minimum bandwidths ($h_{max}$, $h_{min}$). Here, we varied these hyperparameters while keeping all other training settings unchanged to assess their influence on the overall model. Experiments were conducted on a fold (Fold 0) of AACDB dataset (Appendix D), training on 2,939 Ab-Ag complexes and evaluating on 735 complexes, all using the same random seed. The results were reported on the valiadation

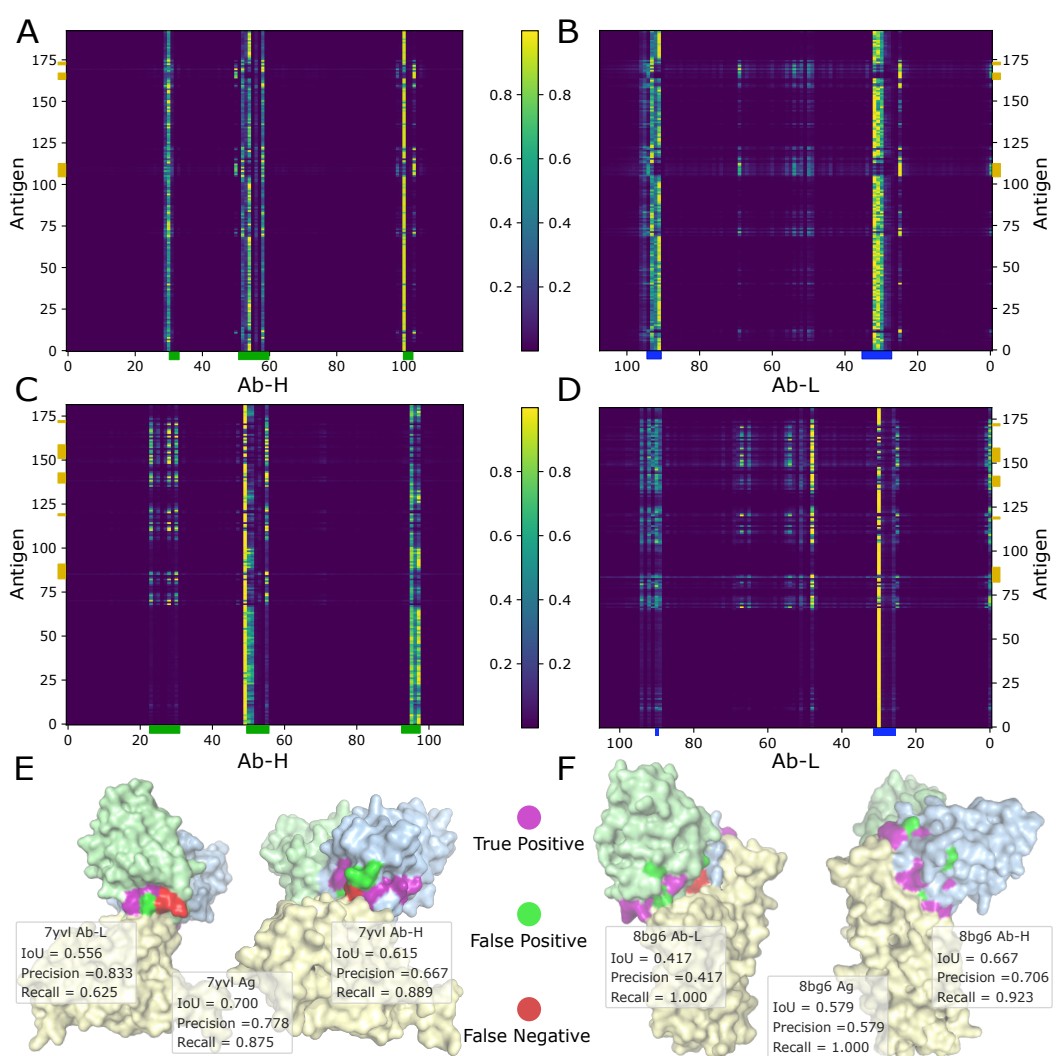

Figure 4: Case study. (A–D) Weighted attention maps from the final sliding step: left, antigen with Ab-H; right, antigen with Ab-L. Color bars attached to the axes indicate the true interface positions. (A,B) 7yvl; (C,D) 8bg6. (E-F) Structural visualization of interface predictions for 7yvl and 8bg6. Surfaces in yellow, blue and green correspond to the antigen, Ab-H and Ab-L, respectively.

set from epochs 40 to 60, which showed the predictive capability near convergence (Appendix E). Additional analyses are provided in Appendix H.

As shown in Figure 3, three key observations can be drawn. First, increasing the number of sliding steps $T$ progressively improves predictive precision, with three iterations showing the best overall performance in our settings. Second, a smaller bandwidth $h$ tends to improve precision by down-weighting the contributions of more distant residues, while reducing recall since these residues may still carry relevant information (Eq. 3). Third, the overall performance shows minor fluctuations across the hyperparameter ranges considered, indicating the robustness of the sliding-attention algorithm in our tasks.

## 3.5 CASE STUDY

To illustrate the interpretability of our model, we selected two examples of weighted attention maps (Eq. 4) and structural visualizations from the SARS-CoV-2 test set. The attention maps show that ABConformer accurately captures three CDRs on VH (*i.e.*, CDR-H1, CDR-H2 and CDR-H3) as

well as CDRs on VL, with strong attention scores concentrated on these regions (Fig. 4A-D). Furthermore, ABConformer highlights antigen residues that are highly related and proximal to the antibody CDRs (Fig. 4A-D). The structural visualizations further demonstrate the high precision and recall achieved by our model in predicting Ab–Ag interfaces (Fig. 4E,F).

# 4 RELATED WORK

**Antibody-specific interface prediction methods.** PECAN integrates graph representation, graph convolution, attention, and transfer learning to model Ab-Ag structural relationships and contextually predict interfaces (Pittala & Bailey-Kellogg, 2020). Honda's work introduces convolution encoders, transformer encoders and a cross-transformer encoder into the backbone, achieving a multi-task model that simultaneously predicts antibody paratopes and antigen epitopes (Honda et al., 2020). Epi-EPMP employs a graph attention network (GAT) with fully connected layers to capture structural cues on antibodies and antigens (Del Vecchio et al., 2021). PeSTo is a parameter-free geometric transformer that directly encodes protein structures as atomic point clouds, using pairwise geometry and multi-head attention to update atom-level scalar and vector states for binding site prediction (Krapp et al., 2023). MIPE uses multi-modal contrastive learning (CL)—intra-modal CL to separate binding and non-binding residues within each modality, and inter-modal CL to align sequence and structure representations—along with multi-head attention layers that compute attention matrices for antibodies and antigens to capture their interaction patterns (Wang et al., 2024). DeepInterAware can evaluate Ab-Ag affinity, identify binding sites, and predict the binding free energy changes due to mutations. Its Interaction Interface-aware Learner (IIL) embeds antigens with ESM-2 and antibodies with AbLang (Olsen et al., 2022), using bilinear attention and convolution blocks to capture interfaces of Ab-Ag complexes (Xia et al., 2025). Epi4Ab encodes antigen sequences with ESM-2 and antibody CDRs with AntiBERTy (Ruffolo et al., 2021), and integrates them with structural features of Ab-Ag into residual interaction graphs, a graph attention network then classifying residues as epitopes, potential epitopes or non-epitopes (Tran et al., 2025).

**Antibody-agnostic epitope prediction methods.** BepiPred-3.0 uses ESM-2 embeddings as input to a feedforward neural network (FFNN) to predict both linear and conformational B-cell epitopes (Clifford et al., 2022). DiscoTope-3.0 uses inverse folding representations from ESM-IF1 (Hsu et al., 2022) and is trained on both predicted and solved structures using a positive-unlabelled ensemble strategy, enabling structure-based B-cell epitope prediction (Høie et al., 2024). SEMA-1D 2.0 adds a fully-connected layer on an ensemble of five ESM-2 models, while SEMA-3D 2.0 follows the same design but replaces ESM-2 with pre-trained Structure-aware Protein language models (SaProt) (Su et al., 2023; Ivanisenko et al., 2024).

# 5 CONCLUSION

In this study, we propose ABConformer, an interface prediction model based on the sliding-attention Conformer architecture. The experimental results highlight three key findings. First, ABConformer demonstrates improvement in several key metrics (*e.g.*, F1 and PCC) for antibody-specific interface prediction and surpasses widely used sequence-based methods in antibody-agnostic epitope prediction. Second, the sliding-attention algorithm considerably improves the precision of antibody-specific epitope prediction while keeping the overall performance at a high level. Third, ABConformer produces interpretable attention maps for antigen–Ab-H and antigen–Ab-L interactions, with feature and spatial attention accurately capturing epitopes and paratopes within the CDRs.

**Future work.** Several avenues remain to be explored. First, previous antibody-specific methods have incorporated antibody embedding techniques such as AntiBERTy (Ruffolo et al., 2021) and AbLang (Olsen et al., 2022); assessing the effectiveness of such embeddings is important for optimizing ABConformer. Second, ABConformer need further evaluation on additional datasets with experimentally resolved structures, and its utility in practical applications requires validation. Third, pan-epitope prediction still leaves substantial room for improvement. Note that in this study, we simply set antibody embeddings to zero to assess the performance of pan-epitope prediction, while this task does not benefit from either the antibody branches or the sliding-attention modules. As future work, we intend to develop a pure Conformer architecture dedicated to antigen sequences, and further examine how convolution and self-attention individually support epitope prediction.

REPRODUCIBILITY STATEMENT

Training details are provided in Appendix E. All code and data will be made publicly available upon acceptance.

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

## STATEMENT OF LLM USAGE

Large Language Models (LLMs) were only used to polish the language of this paper. No LLM was used to generate research ideas, experiments, or analyses.

## A    ANTIBODY-ANTIGEN INTERFACES

**Antibody structure.** A crystal structure of mouse immunoglobulin G (IgG) is shown on the right (Fig. 5), with the paired variable domains at both Fab tips having the ability to interact with specific antigens.

**Interface identification.** Ab-Ag interfaces play a critical role in determining binding specificity and affinity. During immune recognition, epitopes are typically composed of multiple spatially adjacent residues. To capture this interaction, the notions of residue-neighbor and residue-patch were introduced. A residue-neighbor is defined when the minimum distance between heavy atoms of two residues is less than 4 Å, and a residue-patch refers to a group of residues whose atoms lie within 10 Å of a central residue. To identify the interaction residues between the antibody and antigen, we focused on the residue-neighbor relationship, which is given as:

$$\min_{a \in r_i,\, b \in r_j} \|a - b\| < 4\,\text{Å}, \qquad (10)$$

where $r_i$ and $r_j$ are residues from the antigen and antibody chains respectively, and $a$, $b$ represent the heavy atoms within these residues.

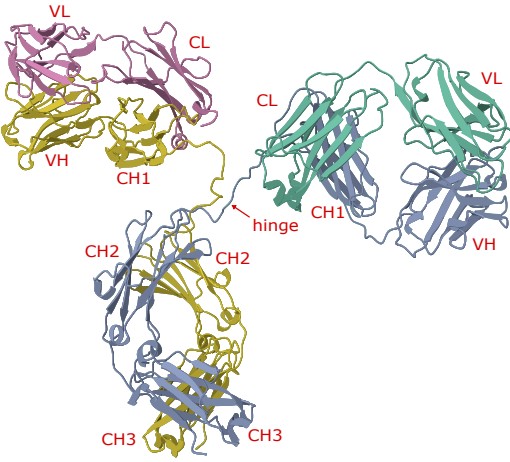

Figure 5: Cartoon representation of a full-length immunoglobulin (PDB ID: 1IGT), with domains annotated. Two identical heavy chains are shown in yellow and blue, and the identical light chains are shown in pink and green.

**CDRs.** Antigen-binding sites are located on the VH and VL domains, where the interacting regions are primarily the CDRs, particularly CDR-H3. The remainder of the variable domain, outside the CDRs, is structurally well conserved and often referred to as the *framework region*. Antibody design is commonly formulated as the task of selecting CDR sequences that optimally conform to a given framework region. In the case study, we show that weighted attention maps of sliding attention accurately capture three CDRs in VH domains, which demonstrates the applicability of our model in identifying functional CDRs.

## B  SLIDING ATTENTION

**Equation 8.** The position update for sliding attention is defined in Equation 7 as

$$P^{(t+1)} = \widehat{W}^{(t)}Q,$$

where $\widehat{W}^{(t)} \in \mathbb{R}^{m \times n}$ is the row-normalized weighted attention matrix, and $Q \in \mathbb{R}^n$ represents the fixed reference sequence positions. Expanding by rows, the update of the $i$-th sliding residue is:

$$p_i^{(t+1)} = \sum_{j=1}^{n} \widehat{W}_{ij}^{(t)} q_j. \tag{11}$$

The displacement from the previous position can be expressed as:

$$p_i^{(t+1)} - p_i^{(t)} = \sum_{j=1}^{n} \widehat{W}_{ij}^{(t)} q_j - p_i^{(t)}. \tag{12}$$

Since $\widehat{W}^{(t)}$ is row-normalized, i.e., $\sum_{j=1}^{n} \widehat{W}_{ij}^{(t)} = 1$, we can factor out $p_i^{(t)}$ to recover Equation 8 in the main text:

$$p_i^{(t+1)} - p_i^{(t)} = \sum_{j=1}^{n} \widehat{W}_{ij}^{(t)} (q_j - p_i^{(t)}).$$

**Algorithm.** Here, an algorithm of sliding attention for a sliding sequence $X$ and reference sequence $Y$ is shown below:

---

**Algorithm 1:** Sliding Attention

---

**Input:** Sliding embeddings $X^{(0)}$, reference embeddings $Y^{(0)}$, initial positions $P^{(0)}$, reference positions $Q$, mask $M$, linear projections $E_S, E_R, E_X, E_Y$, iteration steps $T$, bandwidth constraints $h_{\min}, h_{\max}$, scaling factor $c$, small constant $\varepsilon$.

**Output:** $X^{(T)}, Y^{(T)}, \widehat{W}_{ij}^{(T)}, \widetilde{W}_{ij}^{(T)}$.

01: $h \leftarrow \min\{h_{\max}, \max\{h_{\min}, \sum_{j=1}^{n} M_{:,j}/c\}\}$ (Eq. 3);

02: **for** $t = 0$ **to** $T - 1$ **do**

    03:     // also for all $i \in [1, m]$ and $j \in [1, n]$;

    04:     // feature attention (Eq. 1);

    05:     $a_{ij}^{(t)} \leftarrow (x_i^{(t)} E_S) \cdot (y_j^{(t)} E_R)^\top / \sqrt{d}$;

    06:     $A_{ij}^{(t)} \leftarrow \exp(a_{ij}^{(t)} - \max_k a_{ik}^{(t)})$;

    07:     // spatial attention (Eq. 2);

    08:     $S_{ij}^{(t)} \leftarrow \exp\left(-(p_i^{(t)} - q_j)^2/2h^2\right)$;

    09:     // weighted attention (Eq. 4, 5);

    10:     $W_{ij}^{(t)} \leftarrow M_{ij} (A^{(t)} \odot S^{(t)})_{ij}$;

    11:     $\widehat{W}_{ij}^{(t)} \leftarrow W_{ij}^{(t)}/(\sum_k W_{ik}^{(t)} + \varepsilon)$;

    12:     $\widetilde{W}_{ij}^{(t)} \leftarrow W_{ij}^{(t)}/(\sum_k W_{kj}^{(t)} + \varepsilon)$;

    13:     // Update sliding embeddings and reference embeddings (Eq. 6);

    14:     $X^{(t+1)} \leftarrow \widehat{W}^{(t)}(Y^{(t)} E_Y) + X^{(t)}$;

    15:     $Y^{(t+1)} \leftarrow (\widetilde{W}^{(t)})^\top (X^{(t)} E_X) + Y^{(t)}$;

    16:     // Update sliding positions (Eq. 7);

    17:     $P^{(t+1)} \leftarrow \widehat{W}^{(t)}Q$;

18: **return** $X^{(T)}, Y^{(T)}, \widehat{W}_{ij}^{(T)}, \widetilde{W}_{ij}^{(T)}$.

---

## C  PERFORMANCE METRICS

**Binary predictions.** In the main text, we report intersection over union (IoU), precision (Prec), recall (Rec), F1 score and Matthews correlation coefficient (MCC) for paratope and epitope predictions. These metrics quantify the agreement between predicted and true binding sites after binarization, with higher values indicating better predictive performance:

$$\text{IoU} = \frac{\text{TP}}{\text{TP} + \text{FP} + \text{FN}}$$

$$\text{Prec} = \frac{\text{TP}}{\text{TP} + \text{FP}}$$

$$\text{Rec} = \frac{\text{TP}}{\text{TP} + \text{FN}}$$

$$\text{F1} = 2 \cdot \frac{\text{Prec} \cdot \text{Rec}}{\text{Prec} + \text{Rec}}$$

$$\text{MCC} = \frac{\text{TP} \cdot \text{TN} - \text{FP} \cdot \text{FN}}{\sqrt{(\text{TP} + \text{FP})(\text{TP} + \text{FN})(\text{TN} + \text{FP})(\text{TN} + \text{FN})}}$$

where TP, TN, FP and FN denote true positives, true negatives, false positives and false negatives.

**Score predictions.** Metrics that can be computed from continuous prediction scores include Pearson correlation coefficient (PCC), areas under the receiver operating characteristic (ROC) and precision-recall (PR) curves, Brier score and binary cross-entropy (BCE). These metrics assess the probabilistic calibration and ranking quality of predictions, which are computed as follows:

$$\text{PCC} = \frac{\text{Cov}(y_i, \hat{y}_i)}{\sigma_{y_i} \, \sigma_{\hat{y}_i}}$$

$$\text{ROC-AUC} = \int_0^1 \text{TPR}(t) \, d\,\text{FPR}(t)$$

$$\text{PR-AUC} = \int_0^1 \text{Prec}(t) \, d\,\text{Rec}(t)$$

$$\text{Brier} = \frac{1}{N} \sum_{i=1}^{N} (y_i - \hat{y}_i)^2$$

$$\text{BCE} = -\frac{1}{N} \sum_{i=1}^{N} \left[ y_i \log(\hat{y}_i) + (1 - y_i) \log(1 - \hat{y}_i) \right]$$

Here, $y_i \in \{0, 1\}$ is the true label of residue $i$, $\hat{y}_i \in [0, 1]$ is the predicted score, and $N$ is the total number of residues. The threshold $t \in [0, 1]$ is used to binarize the predicted scores when computing TPR, FPR, Rec and Prec, which are defined as $\text{TPR}(t) = \frac{\text{TP}(t)}{\text{TP}(t) + \text{FN}(t)}, \text{FPR}(t) = \frac{\text{FP}(t)}{\text{FP}(t) + \text{TN}(t)}, \text{Rec}(t) = \frac{\text{TP}(t)}{\text{TP}(t) + \text{FN}(t)}, \text{Prec}(t) = \frac{\text{TP}(t)}{\text{TP}(t) + \text{FP}(t)}.$

# D DATASET

| Fold | Split | Ab-H | | Ab-L | | Ag | |
|------|-------|------|------|------|------|------|------|
| | | Avg. Len. | Int. Rate | Avg. Len. | Int. Rate | Avg. Len. | Int. Rate |
| 0 | Train | 182.1 | 0.073 | 175.9 | 0.051 | 345.7 | 0.077 |
| | Val | 179.6 | 0.076 | 173.0 | 0.053 | 351.8 | 0.075 |
| 1 | Train | 181.8 | 0.074 | 175.3 | 0.052 | 345.6 | 0.076 |
| | Val | 180.9 | 0.072 | 175.5 | 0.051 | 352.4 | 0.077 |
| 2 | Train | 181.2 | 0.074 | 175.0 | 0.052 | 348.4 | 0.077 |
| | Val | 183.0 | 0.073 | 176.6 | 0.050 | 341.2 | 0.075 |
| 3 | Train | 181.7 | 0.073 | 175.5 | 0.051 | 346.9 | 0.076 |
| | Val | 181.2 | 0.074 | 175.0 | 0.052 | 347.0 | 0.077 |
| 4 | Train | 181.2 | 0.074 | 175.0 | 0.052 | 348.1 | 0.076 |
| | Val | 183.2 | 0.073 | 176.8 | 0.051 | 342.4 | 0.078 |

Table 3: Dataset statistics across 5-fold splits. For each fold, we report the average sequence length and the average proportion of interfaces for Ab-H, Ab-L and Ag.

| Fold | Cluster 1 | Cluster 2 | Cluster 3 | Cluster 4 | Cluster 5 | Cluster 6 | SUM |
|------|-----------|-----------|-----------|-----------|-----------|-----------|-----|
| Fold 0 | 156 | 57 | 17 | 220 | 17 | 268 | 735 |
| Fold 1 | 156 | 57 | 16 | 220 | 18 | 268 | 735 |
| Fold 2 | 156 | 57 | 16 | 220 | 18 | 268 | 735 |
| Fold 3 | 155 | 57 | 16 | 221 | 18 | 268 | 735 |
| Fold 4 | 155 | 57 | 16 | 220 | 18 | 268 | 734 |

Table 4: Distribution of validation samples across clusters for each fold.

**AACDB.** The original 7,488 PDB structures were filtered to ensure that each PDB ID appeared only once, resulting in a final set of 3,674 complexes. Antigen sequences were then extracted from these complexes, and a phylogenetic tree of the these sequences was constructed using ClustalOmega. As shown in Figure 6, six clusters were identified based on evolutionary relationships. Each cluster was subsequently divided into five folds, which were then combined to form the final cross-validation datasets, yielding four folds with 735 validation samples (2,939 training samples) and one fold with 734 validation samples (2,940 training samples) (Tab. 4). A detailed analysis of average sequence lengths and average interface proportions for all chains is provided in Table 3.

In practice, structures that do not distinguish Ab-H and Ab-L (*i.e.*, only the full antibody sequence provided) exist. In such cases, we duplicate the chain into both Ab-H and Ab-L to meet the input requirements of our model.

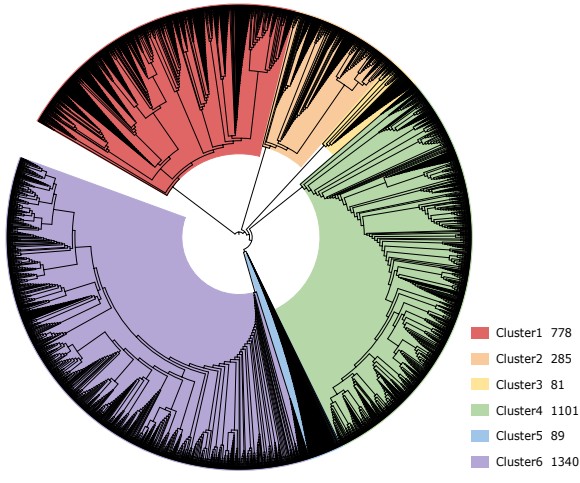

| | | |
|---|---|---|
| Cluster1 | 778 |
| Cluster2 | 285 |
| Cluster3 | 81 |
| Cluster4 | 1101 |
| Cluster5 | 89 |
| Cluster6 | 1340 |

Figure 6: Phylogenetic tree of antigens in the AACDB 3674 dataset, with six clusters obtained.

**SARS-CoV-2.** The SARS-CoV-2 dataset, filtered from CoV-AbDab since 2024, comprises 35 experimentally resolved PDB complexes. Among these, 12 antibodies can target pre-Omicron (SARS-CoV-2 WT and its $\alpha$, $\beta$ variants, etc.), 4 can target Omicron, and 19 have the ability to target both strains. By extracting the Ab-H, Ab-L and corresponding antigen chain from each complex, we obtained 46 entries. This curated small dataset will be made publicly available.

# E    TRAINING AND EVALUATION

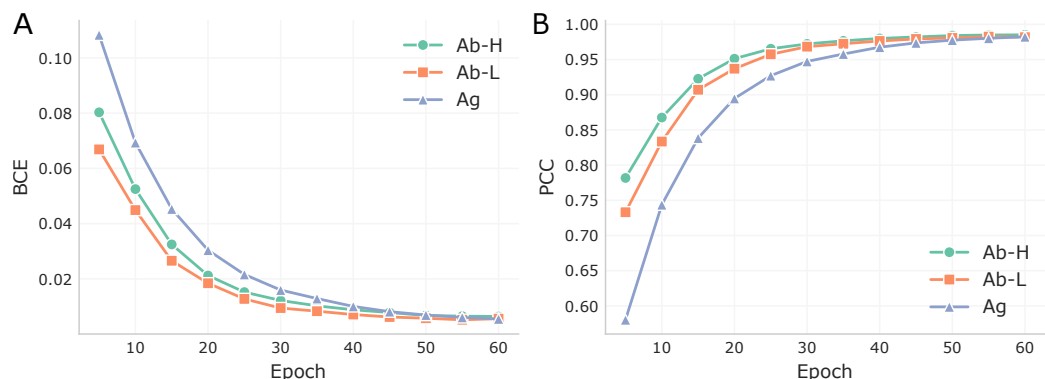

Figure 7: Metrics across training epochs. Metrics were computed on the training set (N=3674) every five epochs for Ab-H, Ab-L and Ag. Each reported value represents the mean calculated over the corresponding epoch together with its two preceding and two succeeding epochs (a five-epoch window). (A) BCE. (B) PCC.

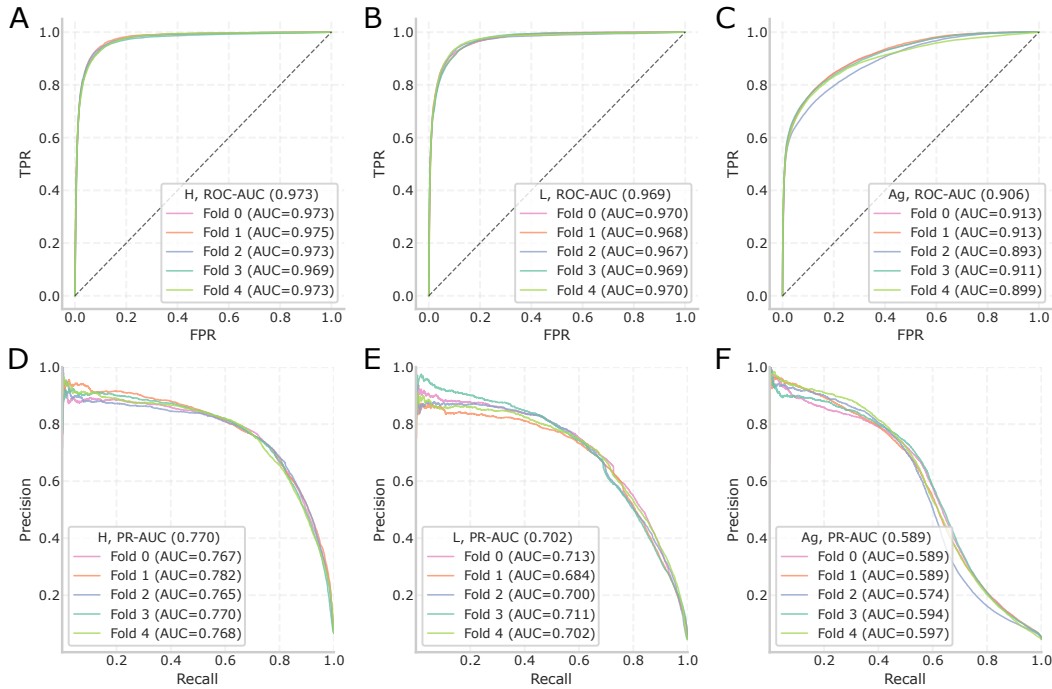

Figure 8: Cross-validation across five folds. The (A-C) ROC-AUC and (D-F) PR-AUC curves are plotted on Ab-H, Ab-L and Ag.

**Training details.** ABConformer was trained using per-residue cross-entropy loss with masking to ignore padded positions. For a batch of sequences, the loss for each chain (Ab-H, Ab-L and Ag) is independently computed as:

$$\mathcal{L}_{\text{chain}} = -\frac{1}{\sum_i m_i} \sum_i m_i \sum_c y_{i,c} \log \hat{y}_{i,c}, \tag{13}$$

where $m_i$ is a binary mask for valid positions, $y_{i,c}$ is the one-hot target for position $i$ and class $c$, and $\hat{y}_{i,c}$ is the predicted probability after softmax. The final loss is averaged across three chains:

| Layer | Epoch | Ab-H | | Ab-L | | Ag | | Param (M) | MACs (G) |
|---|---|---|---|---|---|---|---|---|---|
| | | PCC↑ | F1↑ | PCC↑ | F1↑ | PCC↑ | F1↑ | | |
| 4 | 40-60 | 0.733 | 0.732 | 0.684 | 0.670 | 0.603 | 0.578 | 108.764 | 259.828 |
| 6 | 40-60 | 0.736 | 0.737 | 0.689 | 0.677 | 0.615 | 0.593 | 162.940 | 389.111 |
| 8 | 40-60 | 0.736 | 0.738 | 0.691 | 0.678 | 0.614 | 0.590 | 217.116 | 518.394 |

Table 5: Performance metrics across different layers of Conformer and sliding-attention. Each model was trained on fold 0 (Appendix D) and evaluated on the validation set at epochs 40–60, with the reported values representing the average over these twenty epochs. Params and MACs were calculated assuming a batch size of 2, and all sequences in the batch padded to a length of 512.

$$\mathcal{L} = \frac{1}{3} \left( \mathcal{L}_{H} + \mathcal{L}_{L} + \mathcal{L}_{Ag} \right). \tag{14}$$

Several optimization and stabilization techniques were also employed. First, the model parameters were optimized using AdamW with weight decay, and gradients were clipped to a maximum norm of $1.0$ to prevent instability during backpropagation. Second, to reduce memory usage, we applied automatic mixed precision (AMP). Third, an exponential moving average (EMA) of the model weights was maintained throughout training, improving the stability of evaluation metrics. Finally, the learning rate and loss values were recorded at each iteration using a smoothed logging utility to monitor the optimization process.

The training process of the standard ABConformer (*i.e.*, six layers of stages II and III) on the full AACDB dataset is shown in Figure 7. Predictive performance for Ab-H and Ab-L converges around epoch 40, while Ag converges around epoch 50. This explains our choice of epochs 40–60 in the sensitivity analysis.

**Five-fold cross-validation.** In the ablation studies, all ABConformer variants were evaluated using five-fold cross-validation on the AACDB dataset. Here, we show the five-fold ROC and PR curves for the original ABConformer. As shown in Figure 8, the curves are plotted separately for Ab-H, Ab-L and Ag, indicating similar performance across folds. Notably, epitope prediction performance is consistently lower than that for paratopes. This suggests that the model accurately captures paratope residues within CDRs, and residues between CDRs receive less attention; While antigen binding sites are more variable, making them inherently more challenging for prediction.

**Conformer and sliding-attention layers.** A standard ABConformer consists of six layers of Conformer and sliding-attention modules (Fig. 2). To investigate the effect of model depth, we also explored different numbers of layers. As reported in Table 5, six layers provide the best trade-off between predictive performance and computational cost. Note that in this table, parameter counts (Params) and multiply-accumulate operations (MACs) were calculated using a batch size of 2 and a sequence length of 512. However, during actual training, dynamic sequence length padding was applied for each batch, and a batch size of 6 could be supported in our environment.

**Configuration.** A complete configuration is shown below:

| Parameter | Value | Description |
|---|---|---|
| $d_{model}$ | 640 | Embedding dim of input features. |
| $dim_{ff}$ | 1280 | Hidden dim of feedforward modules. |
| $n_{heads}$ | 10 | Number of attention heads. |
| $conv\_kernel$ | 5 | Kernel size of convolution modules. |
| $n_{blocks}$ | 6 | Number of stacked blocks. |
| $min\_bw$ | 48 | Minimum bandwidth. |
| $max\_bw$ | 144 | Maximum bandwidth. |
| $scale$ | 3 | Scaling factor for the bandwidth. |
| $sliding\_step$ | 3 | Number of sliding steps. |
| $\alpha$ | 0.5 | Weight for Ag update from Ab-H. |

Table 6: Model configuration.

| Env | Spec |
|---|---|
| OS | Linux 5.10.0-35 |
| Python | 3.9.23 |
| CPU | 24C / 48T |
| Memory | 334 GB |
| GPU | $4 \times$ A100 (40GB) |

Table 7: Environment configuration.

## F    COMPARISON EXPERIMENTS

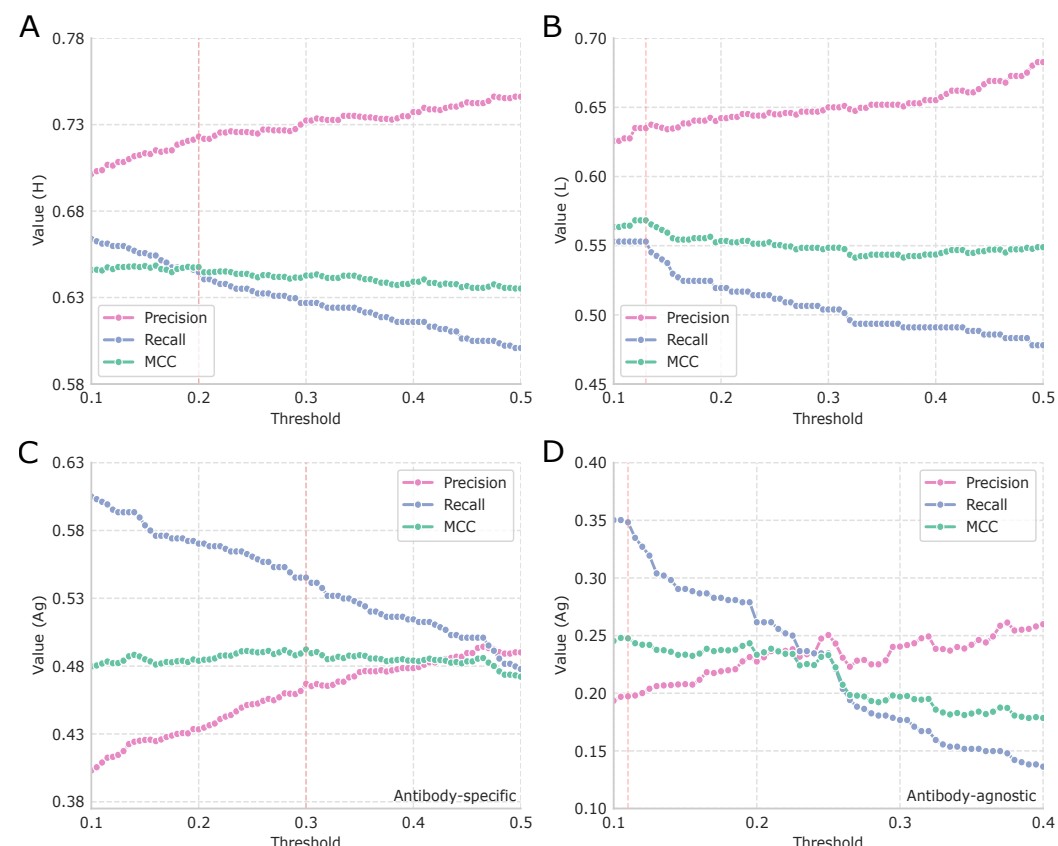

Figure 9: Metrics across thresholds evaluated on the SARS-CoV-2 dataset. The light-red dashed lines indicate the thresholds selected in the comparison experiments. (A–C) Antibody-specific interface prediction on Ab-H, Ab-L and Ag, respectively. (D) Antibody-agnostic epitope prediction on Ag.

**Alphafold Multimer v3.** We used Alphafold Multimer ColabFold v3 with 4 seeds, 5 models and 3 recycles, generating 20 structures per complex. Then we selected the top-ranked predicted structure for each complex and extracted interfaces based on a 4 Å distance cutoff.

**Antibody-specific methods.** Open-source implementations of PECAN, Epi-EPMP, PeSTo, DeepInterAware and Epi4Ab are available on GitHub. PECAN, DeepInterAware and Epi4Ab were trained on the AACDB-3674 dataset (with Epi4Ab operating on Ab-H and Ab-L seperately) and then evaluated on the SARS-CoV-2 dataset. For PeSTo, a parameter-free method with detailed usage guidelines, predictions were obtained by following the provided instructions. Epi-EPMP lacks detailed training code; therefore, we reconstructed the network following their paper and conducted the analysis. For the remaining baselines, we re-implemented their architectures following the descriptions in the original publications.

**Antibody-agnostic methods.** BepiPred-3.0, DiscoTope-3.0 and SEMA 2.0 provide publicly available web platforms for direct use. SEMA-1D 2.0 adopts a 12 Å distance cutoff, achieving the highest recall (Tab. 1). SEMA-3D 2.0 provides a log-scaled score representing the expected number of contacts with antibody residues and annotations of predicted N-glycosylation sites, the interpretation of this score as a probability remains unclear. Hence, we did not report its results.

**ABConformer.** Figure 9 shows metrics evaluated across different thresholds on the SARS-CoV-2 dataset. The thresholds selected for classifying interfaces are 0.2, 0.13 and 0.3 for Ab-H, Ab-L and Ag, respectively, and a threshold of 0.11 was chosen for antibody-agnostic epitope prediction.

# G ABLATION

| Ab | Ab-H | | | Ab-L | | | Ag | | | Param | MACs |
|---|---|---|---|---|---|---|---|---|---|---|---|
| MHSA | F1↑ | PCC↑ | PR↑ | F1↑ | PCC↑ | PR↑ | F1↑ | PCC↑ | PR↑ | (M) | (G) |
| ✗ | 0.738 | 0.741 | 0.770 | 0.679 | 0.697 | 0.702 | 0.597 | 0.611 | 0.589 | 162.940 | 389.111 |
| ✓ | 0.741 | 0.743 | 0.770 | 0.677 | 0.701 | 0.705 | 0.591 | 0.608 | 0.585 | 167.862 | 394.144 |

Table 8: Ablation study of MHSA modules in antibody branches. Mean metrics of 5-fold cross-validation were evaluated on the AACDB dataset.

| ABConformer | | IoU↑ | Prec↑ | Rec↑ | F1↑ | MCC↑ | PCC↑ | ROC↑ | PR↑ | Brier↓ | BCE↓ |
|---|---|---|---|---|---|---|---|---|---|---|---|
| Ab-H | fold 0 | 0.588 | 0.742 | 0.740 | 0.741 | 0.722 | 0.741 | 0.973 | 0.767 | 0.031 | 0.200 |
| | fold 1 | 0.589 | 0.717 | 0.768 | 0.741 | 0.723 | 0.743 | 0.975 | 0.782 | 0.030 | 0.185 |
| | fold 2 | 0.587 | 0.696 | 0.789 | 0.739 | 0.721 | 0.746 | 0.973 | 0.765 | 0.031 | 0.192 |
| | fold 3 | 0.587 | 0.719 | 0.761 | 0.739 | 0.720 | 0.742 | 0.969 | 0.770 | 0.031 | 0.197 |
| | fold 4 | 0.575 | 0.763 | 0.701 | 0.731 | 0.713 | 0.732 | 0.973 | 0.768 | 0.031 | 0.211 |
| | **AVG** | **0.585** | **0.727** | **0.752** | **0.738** | **0.720** | **0.741** | **0.973** | **0.770** | **0.031** | **0.197** |
| Ab-L | fold 0 | 0.520 | 0.703 | 0.666 | 0.684 | 0.669 | 0.695 | 0.970 | 0.713 | 0.025 | 0.150 |
| | fold 1 | 0.513 | 0.676 | 0.680 | 0.678 | 0.662 | 0.697 | 0.968 | 0.684 | 0.026 | 0.164 |
| | fold 2 | 0.513 | 0.710 | 0.649 | 0.678 | 0.665 | 0.698 | 0.967 | 0.700 | 0.024 | 0.162 |
| | fold 3 | 0.506 | 0.718 | 0.632 | 0.672 | 0.659 | 0.694 | 0.969 | 0.711 | 0.025 | 0.166 |
| | fold 4 | 0.521 | 0.687 | 0.684 | 0.685 | 0.671 | 0.701 | 0.970 | 0.702 | 0.025 | 0.172 |
| | **AVG** | **0.514** | **0.699** | **0.662** | **0.679** | **0.665** | **0.697** | **0.969** | **0.702** | **0.025** | **0.163** |
| Ag | fold 0 | 0.432 | 0.639 | 0.571 | 0.603 | 0.586 | 0.616 | 0.913 | 0.589 | 0.030 | 0.206 |
| | fold 1 | 0.420 | 0.673 | 0.528 | 0.592 | 0.579 | 0.605 | 0.913 | 0.589 | 0.029 | 0.231 |
| | fold 2 | 0.415 | 0.644 | 0.539 | 0.587 | 0.571 | 0.600 | 0.893 | 0.574 | 0.031 | 0.232 |
| | fold 3 | 0.437 | 0.674 | 0.555 | 0.609 | 0.594 | 0.622 | 0.911 | 0.594 | 0.029 | 0.217 |
| | fold 4 | 0.424 | 0.668 | 0.538 | 0.596 | 0.582 | 0.610 | 0.899 | 0.597 | 0.032 | 0.230 |
| | **AVG** | **0.426** | **0.660** | **0.546** | **0.597** | **0.583** | **0.611** | **0.906** | **0.589** | **0.030** | **0.223** |

| I: one-hot | | IoU↑ | Prec↑ | Rec↑ | F1↑ | MCC↑ | PCC↑ | ROC↑ | PR↑ | Brier↓ | BCE↓ |
|---|---|---|---|---|---|---|---|---|---|---|---|
| Ab-H | fold 0 | 0.572 | 0.687 | 0.774 | 0.728 | 0.708 | 0.741 | 0.974 | 0.760 | 0.031 | 0.134 |
| | fold 1 | 0.571 | 0.705 | 0.751 | 0.727 | 0.707 | 0.738 | 0.970 | 0.761 | 0.031 | 0.147 |
| | fold 2 | 0.570 | 0.707 | 0.746 | 0.726 | 0.705 | 0.733 | 0.973 | 0.757 | 0.032 | 0.154 |
| | fold 3 | 0.569 | 0.716 | 0.734 | 0.725 | 0.705 | 0.732 | 0.961 | 0.739 | 0.031 | 0.185 |
| | fold 4 | 0.572 | 0.691 | 0.768 | 0.728 | 0.708 | 0.739 | 0.971 | 0.757 | 0.030 | 0.146 |
| | **AVG** | **0.571** | **0.701** | **0.755** | **0.727** | **0.707** | **0.737** | **0.970** | **0.755** | **0.031** | **0.153** |
| Ab-L | fold 0 | 0.502 | 0.651 | 0.687 | 0.669 | 0.653 | 0.693 | 0.967 | 0.688 | 0.024 | 0.104 |
| | fold 1 | 0.510 | 0.672 | 0.680 | 0.676 | 0.660 | 0.694 | 0.964 | 0.695 | 0.025 | 0.122 |
| | fold 2 | 0.505 | 0.632 | 0.716 | 0.671 | 0.655 | 0.691 | 0.967 | 0.690 | 0.026 | 0.118 |
| | fold 3 | 0.511 | 0.664 | 0.689 | 0.676 | 0.661 | 0.689 | 0.951 | 0.683 | 0.024 | 0.158 |
| | fold 4 | 0.502 | 0.675 | 0.663 | 0.669 | 0.653 | 0.687 | 0.964 | 0.699 | 0.023 | 0.127 |
| | **AVG** | **0.506** | **0.659** | **0.687** | **0.672** | **0.657** | **0.691** | **0.963** | **0.691** | **0.025** | **0.126** |
| Ag | fold 0 | 0.324 | 0.486 | 0.492 | 0.489 | 0.464 | 0.536 | 0.896 | 0.507 | 0.033 | 0.141 |
| | fold 1 | 0.319 | 0.535 | 0.441 | 0.483 | 0.463 | 0.536 | 0.891 | 0.507 | 0.034 | 0.164 |
| | fold 2 | 0.294 | 0.447 | 0.461 | 0.454 | 0.427 | 0.496 | 0.886 | 0.453 | 0.036 | 0.150 |
| | fold 3 | 0.362 | 0.514 | 0.550 | 0.532 | 0.508 | 0.559 | 0.890 | 0.520 | 0.035 | 0.188 |
| | fold 4 | 0.341 | 0.511 | 0.507 | 0.509 | 0.485 | 0.551 | 0.898 | 0.526 | 0.034 | 0.149 |
| | **AVG** | **0.328** | **0.499** | **0.490** | **0.493** | **0.470** | **0.536** | **0.892** | **0.502** | **0.034** | **0.158** |

| III: cross-att | | IoU↑ | Prec↑ | Rec↑ | F1↑ | MCC↑ | PCC↑ | ROC↑ | PR↑ | Brier↓ | BCE↓ |
|---|---|---|---|---|---|---|---|---|---|---|---|
| Ab-H | fold 0 | 0.589 | 0.722 | 0.761 | 0.741 | 0.722 | 0.744 | 0.972 | 0.769 | 0.032 | 0.197 |
| | fold 1 | 0.588 | 0.717 | 0.766 | 0.741 | 0.722 | 0.744 | 0.972 | 0.767 | 0.031 | 0.204 |
| | fold 2 | 0.578 | 0.741 | 0.725 | 0.733 | 0.714 | 0.734 | 0.971 | 0.759 | 0.031 | 0.207 |
| | fold 3 | 0.574 | 0.708 | 0.753 | 0.730 | 0.710 | 0.734 | 0.968 | 0.748 | 0.032 | 0.191 |
| | fold 4 | 0.586 | 0.714 | 0.765 | 0.739 | 0.720 | 0.740 | 0.974 | 0.762 | 0.031 | 0.208 |
| | **AVG** | **0.583** | **0.721** | **0.754** | **0.737** | **0.718** | **0.739** | **0.971** | **0.761** | **0.031** | **0.201** |
| Ab-L | fold 0 | 0.520 | 0.720 | 0.651 | 0.684 | 0.670 | 0.690 | 0.974 | 0.718 | 0.025 | 0.168 |
| | fold 1 | 0.509 | 0.641 | 0.713 | 0.675 | 0.659 | 0.691 | 0.969 | 0.703 | 0.026 | 0.146 |
| | fold 2 | 0.512 | 0.682 | 0.672 | 0.677 | 0.662 | 0.685 | 0.969 | 0.700 | 0.025 | 0.169 |
| | fold 3 | 0.520 | 0.685 | 0.683 | 0.684 | 0.669 | 0.693 | 0.964 | 0.688 | 0.025 | 0.163 |
| | fold 4 | 0.518 | 0.647 | 0.721 | 0.682 | 0.667 | 0.695 | 0.972 | 0.698 | 0.025 | 0.148 |
| | **AVG** | **0.516** | **0.675** | **0.688** | **0.680** | **0.666** | **0.691** | **0.970** | **0.701** | **0.025** | **0.159** |

| | | IoU ↑ | Prec ↑ | Rec ↑ | F1 ↑ | MCC ↑ | PCC ↑ | ROC ↑ | PR ↑ | Brier ↓ | BCE ↓ |
|---|---|---|---|---|---|---|---|---|---|---|---|
| Ag | fold 0 | 0.401 | 0.552 | 0.594 | 0.572 | 0.551 | 0.595 | 0.912 | 0.554 | 0.033 | 0.215 |
| | fold 1 | 0.395 | 0.541 | 0.595 | 0.567 | 0.546 | 0.578 | 0.916 | 0.578 | 0.035 | 0.221 |
| | fold 2 | 0.385 | 0.543 | 0.570 | 0.556 | 0.534 | 0.575 | 0.903 | 0.575 | 0.035 | 0.217 |
| | fold 3 | 0.399 | 0.553 | 0.590 | 0.571 | 0.550 | 0.584 | 0.896 | 0.544 | 0.034 | 0.219 |
| | fold 4 | 0.386 | 0.527 | 0.591 | 0.557 | 0.535 | 0.572 | 0.891 | 0.556 | 0.032 | 0.224 |
| | **AVG** | **0.393** | **0.543** | **0.588** | **0.565** | **0.543** | **0.581** | **0.903** | **0.562** | **0.034** | **0.219** |

| *III: self-att* | | IoU ↑ | Prec ↑ | Rec ↑ | F1 ↑ | MCC ↑ | PCC ↑ | ROC ↑ | PR ↑ | Brier ↓ | BCE ↓ |
|---|---|---|---|---|---|---|---|---|---|---|---|
| Ab-H | fold 0 | 0.587 | 0.716 | 0.765 | 0.740 | 0.720 | 0.741 | 0.972 | 0.771 | 0.032 | 0.218 |
| | fold 1 | 0.575 | 0.719 | 0.741 | 0.730 | 0.711 | 0.730 | 0.971 | 0.743 | 0.030 | 0.192 |
| | fold 2 | 0.577 | 0.713 | 0.751 | 0.731 | 0.712 | 0.729 | 0.970 | 0.740 | 0.033 | 0.212 |
| | fold 3 | 0.587 | 0.738 | 0.742 | 0.740 | 0.721 | 0.739 | 0.971 | 0.774 | 0.031 | 0.241 |
| | fold 4 | 0.588 | 0.720 | 0.763 | 0.741 | 0.722 | 0.741 | 0.971 | 0.769 | 0.032 | 0.224 |
| | **AVG** | **0.583** | **0.721** | **0.752** | **0.736** | **0.717** | **0.736** | **0.971** | **0.759** | **0.032** | **0.217** |
| Ab-L | fold 0 | 0.517 | 0.663 | 0.702 | 0.682 | 0.666 | 0.691 | 0.961 | 0.696 | 0.026 | 0.183 |
| | fold 1 | 0.482 | 0.629 | 0.673 | 0.650 | 0.633 | 0.657 | 0.945 | 0.638 | 0.028 | 0.189 |
| | fold 2 | 0.485 | 0.643 | 0.664 | 0.653 | 0.636 | 0.657 | 0.948 | 0.633 | 0.028 | 0.201 |
| | fold 3 | 0.518 | 0.687 | 0.678 | 0.682 | 0.667 | 0.689 | 0.962 | 0.686 | 0.025 | 0.179 |
| | fold 4 | 0.508 | 0.684 | 0.664 | 0.673 | 0.658 | 0.682 | 0.950 | 0.674 | 0.026 | 0.178 |
| | **AVG** | **0.502** | **0.661** | **0.676** | **0.668** | **0.652** | **0.675** | **0.953** | **0.665** | **0.026** | **0.186** |
| Ag | fold 0 | 0.310 | 0.485 | 0.462 | 0.473 | 0.449 | 0.495 | 0.875 | 0.425 | 0.037 | 0.245 |
| | fold 1 | 0.292 | 0.431 | 0.475 | 0.452 | 0.425 | 0.466 | 0.874 | 0.393 | 0.039 | 0.239 |
| | fold 2 | 0.288 | 0.462 | 0.434 | 0.448 | 0.422 | 0.482 | 0.875 | 0.414 | 0.034 | 0.214 |
| | fold 3 | 0.302 | 0.501 | 0.431 | 0.464 | 0.441 | 0.487 | 0.879 | 0.419 | 0.036 | 0.231 |
| | fold 4 | 0.303 | 0.467 | 0.463 | 0.465 | 0.440 | 0.495 | 0.882 | 0.425 | 0.036 | 0.231 |
| | **AVG** | **0.299** | **0.469** | **0.453** | **0.460** | **0.435** | **0.485** | **0.877** | **0.415** | **0.037** | **0.232** |

| *II: no conv* | | IoU ↑ | Prec ↑ | Rec ↑ | F1 ↑ | MCC ↑ | PCC ↑ | ROC ↑ | PR ↑ | Brier ↓ | BCE ↓ |
|---|---|---|---|---|---|---|---|---|---|---|---|
| Ab-H | fold 0 | 0.565 | 0.700 | 0.746 | 0.722 | 0.701 | 0.733 | 0.970 | 0.749 | 0.032 | 0.163 |
| | fold 1 | 0.567 | 0.720 | 0.728 | 0.724 | 0.704 | 0.734 | 0.971 | 0.760 | 0.030 | 0.166 |
| | fold 2 | 0.566 | 0.720 | 0.726 | 0.723 | 0.703 | 0.730 | 0.968 | 0.738 | 0.031 | 0.163 |
| | fold 3 | 0.578 | 0.709 | 0.757 | 0.732 | 0.713 | 0.739 | 0.964 | 0.739 | 0.031 | 0.170 |
| | fold 4 | 0.579 | 0.713 | 0.755 | 0.734 | 0.714 | 0.739 | 0.970 | 0.748 | 0.031 | 0.168 |
| | **AVG** | **0.571** | **0.713** | **0.742** | **0.727** | **0.707** | **0.735** | **0.969** | **0.747** | **0.031** | **0.166** |
| Ab-L | fold 0 | 0.516 | 0.690 | 0.673 | 0.681 | 0.666 | 0.694 | 0.968 | 0.692 | 0.024 | 0.138 |
| | fold 1 | 0.498 | 0.653 | 0.678 | 0.665 | 0.649 | 0.678 | 0.960 | 0.679 | 0.026 | 0.167 |
| | fold 2 | 0.505 | 0.663 | 0.679 | 0.671 | 0.656 | 0.685 | 0.961 | 0.662 | 0.024 | 0.143 |
| | fold 3 | 0.500 | 0.631 | 0.706 | 0.666 | 0.651 | 0.687 | 0.964 | 0.672 | 0.026 | 0.138 |
| | fold 4 | 0.512 | 0.671 | 0.682 | 0.677 | 0.662 | 0.691 | 0.963 | 0.677 | 0.024 | 0.139 |
| | **AVG** | **0.506** | **0.662** | **0.684** | **0.672** | **0.657** | **0.687** | **0.963** | **0.676** | **0.025** | **0.145** |
| Ag | fold 0 | 0.379 | 0.547 | 0.553 | 0.550 | 0.528 | 0.580 | 0.906 | 0.551 | 0.032 | 0.181 |
| | fold 1 | 0.379 | 0.560 | 0.539 | 0.549 | 0.529 | 0.572 | 0.895 | 0.534 | 0.032 | 0.195 |
| | fold 2 | 0.367 | 0.560 | 0.517 | 0.537 | 0.516 | 0.558 | 0.900 | 0.518 | 0.034 | 0.210 |
| | fold 3 | 0.398 | 0.607 | 0.537 | 0.570 | 0.551 | 0.599 | 0.904 | 0.575 | 0.030 | 0.192 |
| | fold 4 | 0.359 | 0.510 | 0.548 | 0.529 | 0.505 | 0.552 | 0.898 | 0.516 | 0.036 | 0.215 |
| | **AVG** | **0.377** | **0.557** | **0.539** | **0.547** | **0.526** | **0.572** | **0.901** | **0.539** | **0.033** | **0.198** |

| *II: no MHSA* | | IoU ↑ | Prec ↑ | Rec ↑ | F1 ↑ | MCC ↑ | PCC ↑ | ROC ↑ | PR ↑ | Brier ↓ | BCE ↓ |
|---|---|---|---|---|---|---|---|---|---|---|---|
| Ab-H | fold 0 | 0.590 | 0.736 | 0.748 | 0.742 | 0.723 | 0.743 | 0.974 | 0.781 | 0.031 | 0.192 |
| | fold 1 | 0.578 | 0.738 | 0.728 | 0.733 | 0.714 | 0.735 | 0.970 | 0.762 | 0.031 | 0.199 |
| | fold 2 | 0.578 | 0.709 | 0.757 | 0.732 | 0.713 | 0.739 | 0.964 | 0.739 | 0.031 | 0.170 |
| | fold 3 | 0.583 | 0.742 | 0.731 | 0.737 | 0.718 | 0.739 | 0.970 | 0.778 | 0.031 | 0.200 |
| | fold 4 | 0.589 | 0.704 | 0.784 | 0.741 | 0.723 | 0.740 | 0.969 | 0.735 | 0.031 | 0.206 |
| | **AVG** | **0.584** | **0.726** | **0.749** | **0.737** | **0.718** | **0.739** | **0.970** | **0.759** | **0.031** | **0.193** |
| Ab-L | fold 0 | 0.508 | 0.667 | 0.681 | 0.674 | 0.658 | 0.691 | 0.960 | 0.673 | 0.027 | 0.167 |
| | fold 1 | 0.522 | 0.640 | 0.738 | 0.686 | 0.672 | 0.695 | 0.973 | 0.699 | 0.026 | 0.162 |
| | fold 2 | 0.500 | 0.631 | 0.706 | 0.666 | 0.651 | 0.693 | 0.964 | 0.672 | 0.026 | 0.148 |
| | fold 3 | 0.512 | 0.674 | 0.681 | 0.678 | 0.662 | 0.694 | 0.969 | 0.711 | 0.025 | 0.153 |
| | fold 4 | 0.511 | 0.670 | 0.683 | 0.676 | 0.661 | 0.693 | 0.968 | 0.672 | 0.025 | 0.165 |
| | **AVG** | **0.511** | **0.656** | **0.698** | **0.676** | **0.661** | **0.693** | **0.967** | **0.685** | **0.026** | **0.159** |
| Ag | fold 0 | 0.415 | 0.609 | 0.566 | 0.587 | 0.568 | 0.600 | 0.903 | 0.580 | 0.031 | 0.219 |
| | fold 1 | 0.401 | 0.571 | 0.574 | 0.572 | 0.551 | 0.581 | 0.908 | 0.569 | 0.034 | 0.244 |
| | fold 2 | 0.398 | 0.607 | 0.537 | 0.570 | 0.551 | 0.599 | 0.904 | 0.575 | 0.030 | 0.212 |
| | fold 3 | 0.422 | 0.604 | 0.583 | 0.593 | 0.574 | 0.604 | 0.917 | 0.593 | 0.032 | 0.224 |
| | fold 4 | 0.419 | 0.659 | 0.534 | 0.590 | 0.575 | 0.601 | 0.894 | 0.565 | 0.031 | 0.243 |
| | **AVG** | **0.411** | **0.610** | **0.559** | **0.582** | **0.564** | **0.597** | **0.905** | **0.576** | **0.032** | **0.229** |

| *II, III: no att* | | IoU↑ | Prec↑ | Rec↑ | F1↑ | MCC↑ | PCC↑ | ROC↑ | PR↑ | Brier↓ | BCE↓ |
|---|---|---|---|---|---|---|---|---|---|---|---|
| Ab-H | fold 0 | 0.572 | 0.705 | 0.753 | 0.728 | 0.708 | 0.732 | 0.946 | 0.730 | 0.033 | 0.221 |
| | fold 1 | 0.578 | 0.706 | 0.761 | 0.732 | 0.713 | 0.735 | 0.956 | 0.734 | 0.031 | 0.215 |
| | fold 2 | 0.575 | 0.705 | 0.757 | 0.730 | 0.711 | 0.734 | 0.958 | 0.730 | 0.031 | 0.196 |
| | fold 3 | 0.564 | 0.704 | 0.740 | 0.721 | 0.701 | 0.728 | 0.957 | 0.722 | 0.032 | 0.204 |
| | fold 4 | 0.571 | 0.711 | 0.743 | 0.727 | 0.707 | 0.733 | 0.951 | 0.724 | 0.031 | 0.199 |
| | **AVG** | **0.572** | **0.706** | **0.751** | **0.728** | **0.708** | **0.732** | **0.954** | **0.728** | **0.032** | **0.207** |
| Ab-L | fold 0 | 0.493 | 0.638 | 0.684 | 0.660 | 0.644 | 0.676 | 0.932 | 0.639 | 0.026 | 0.165 |
| | fold 1 | 0.473 | 0.611 | 0.676 | 0.642 | 0.625 | 0.661 | 0.927 | 0.623 | 0.027 | 0.178 |
| | fold 2 | 0.484 | 0.647 | 0.657 | 0.652 | 0.636 | 0.674 | 0.929 | 0.654 | 0.025 | 0.148 |
| | fold 3 | 0.471 | 0.624 | 0.658 | 0.640 | 0.623 | 0.660 | 0.927 | 0.624 | 0.026 | 0.157 |
| | fold 4 | 0.477 | 0.652 | 0.640 | 0.646 | 0.629 | 0.666 | 0.930 | 0.633 | 0.025 | 0.154 |
| | **AVG** | **0.479** | **0.635** | **0.663** | **0.648** | **0.631** | **0.667** | **0.929** | **0.635** | **0.026** | **0.161** |
| Ag | fold 0 | 0.302 | 0.475 | 0.453 | 0.464 | 0.439 | 0.490 | 0.864 | 0.417 | 0.036 | 0.222 |
| | fold 1 | 0.287 | 0.436 | 0.457 | 0.446 | 0.420 | 0.479 | 0.860 | 0.403 | 0.036 | 0.218 |
| | fold 2 | 0.283 | 0.438 | 0.444 | 0.441 | 0.413 | 0.474 | 0.861 | 0.398 | 0.038 | 0.225 |
| | fold 3 | 0.300 | 0.483 | 0.443 | 0.462 | 0.437 | 0.492 | 0.850 | 0.418 | 0.036 | 0.246 |
| | fold 4 | 0.294 | 0.471 | 0.439 | 0.454 | 0.429 | 0.487 | 0.863 | 0.420 | 0.037 | 0.234 |
| | **AVG** | **0.293** | **0.460** | **0.447** | **0.453** | **0.427** | **0.484** | **0.859** | **0.411** | **0.037** | **0.229** |

Table 9: Details of Ablation Studies. Performance of interface prediction was evaluated on Ab-H, Ab-L and Ag using five-fold cross-validation. AACDB (N=3,674; four folds with 735 validation complexes, one with 734). *Threshold: 0.33*.

**Ablation of ABConformer.** Initially, we designed ABConformer by integrating sliding attention into the Conformer architecture. However, the ablation of MHSA modules on the antibody branches indicates that MHSA contributes little to paratope prediction while increasing the computational cost (Tab. 8). The complete results of the ablation studies are shown in Table 9.

# H  SENSITIVITY ANALYSIS

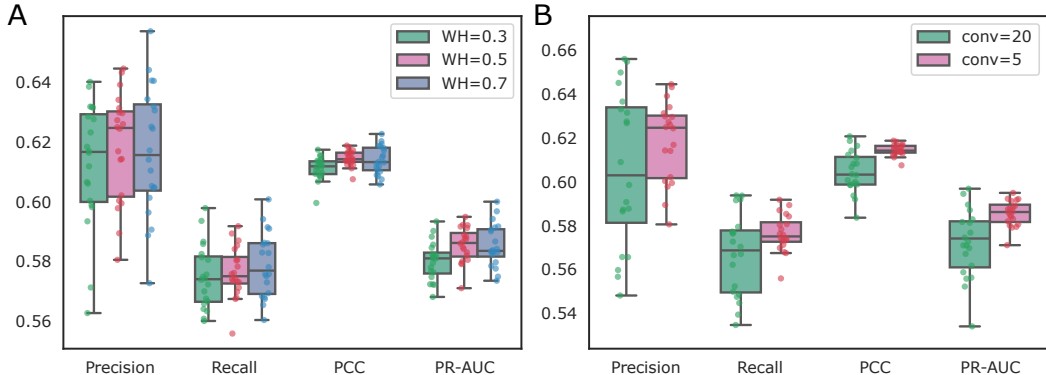

Figure 10: Sensitivity analysis on (A) $\alpha$ (Eq. 9) and (B) convolution kernel.

Here, we further analyzed the weights for updating antigen embeddings after sliding with Ab-H and Ab-L, as well as the convolution kernel in the model (Fig. 10). The results indicate that biasing the weight toward Ab-H or Ab-L reduces epitope precision, while weights above 0.5 (favoring Ab-H) slightly improve recall. Additionally, large convolution kernels tend to overlook fine-grained features within interaction sites, thus decreasing overall performance.

## I   MORE CASES

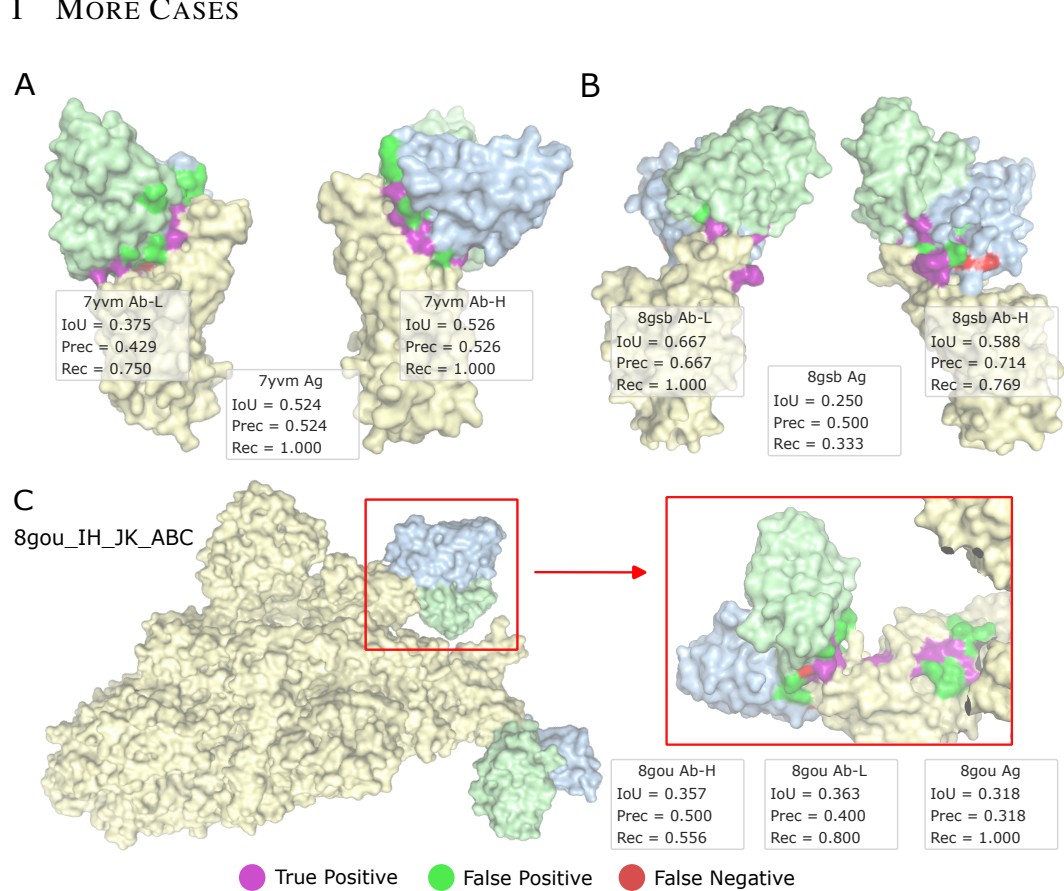

Figure 11: More cases in the SARS-CoV-2 dataset. Surfaces colored in yellow, blue and green represent the antigen, Ab-H and Ab-L, repectively. (A) 7yvm. (B) 8gsb. (C) 8gou. The other pair of antibody chains in 8gou was hidden in the subfigure.

Additional SARS-CoV-2 prediction cases are shown in Figure 11. We further analyzed a complex containing multiple antibody chains (*i.e.*, two paired VH and VL domains) bound to the SARS-CoV-2 Omicron spike protein (PDB ID: 8gou). Since ABConformer requires only one Ab-H, Ab-L and the antigen as input, it additionally predicts pan-epitopes on all possible regions of the antigen. Notably, these pan-epitope predictions coincide with the true binding sites of the other antibody chains, highlighting the potential of our model to generalize to more complex Ab–Ag assemblies.

