# OpenReview forum: "ABConformer: Physics‑inspired Sliding Attention for Antibody-Antigen Interface Prediction"
_ICLR.cc/2026/Conference — Submitted to ICLR 2026_

### Official Review · Reviewer_r9ov · 2025-10-28

**Soundness:** 1
**Presentation:** 1
**Contribution:** 1
**Rating:** 0
**Confidence:** 5

**Summary:**

In this paper, the authors presented ABConformer, a physics-inspired sliding attention scheme to predict paratopes and epitopes given the antibody and antigen sequence. The method is evaluated on the SARS-CoV-2 Ab-Ag dataset against other sequence-based methods.

**Strengths:**

It is very hard to name a strength of the paper, see my comments in the weaknesses part.

**Weaknesses:**

# **This paper is basically copying things from a prior publication in Nature Machine Intelligence.**

This paper is obviously following the paper *Sliding-attention transformer neural architecture for predicting T cell receptor-antigen-human leucocyte antigen binding, Nature Machine Intelligence, 2024, 6(10): 1216-1230*. All the key technical details are similar, including the definition of the distance based attention and embedding based attention, how they are combined together with a template M, and how the sliding attention updates the 1-D coordinate of the protein sequences. The titles of the two papers are also identical. The only difference is the datasets used.

While drawing inspiration from published work is common, outright copying ideas and algorithms from a journal paper and presenting them as your own at a conference crosses ethical boundaries and undermines academic integrity.

**Questions:**

I am curious about what led the authors to believe that such a blatant plagiarism would escape detection by reviewers?

**Details Of Ethics Concerns:**

I am quite concerned that the paper is copying things from a published journal paper (see my comments above).

---

> ### Author Response · Authors · 2025-11-12
>
> We thank the reviewer for noticing the relation to Feng et al. [1]. Our study was developed independently and explicitly cites this prior work in both the *Introduction* and *Methods*. While our model shares the general idea of sliding attention, its algorithmic framework, research focus and experimental results are distinct, as detailed below.
>
> **First, regarding the reference to the prior work.** In the *Introduction*, where the concept of sliding attention is first introduced, we explicitly cite this work and never claim to be the first to propose such an idea (line 88). Furthermore, in the opening paragraph of *Section 2.1*, we cite this reference again to clearly acknowledge the source of inspiration for our idea and algorithmic formulation (line 107).
>
> **Second, regarding the algorithmic design.** In *Section 2.1*, we follow the general idea of constructing feature and spatial attention as well as the sliding update process, but introduce dynamic bandwidth (Eq.3) and additional normalization strategies to handle variable antibody–antigen lengths. Another major difference from Feng et al. is that our sliding attention updates feature embeddings step-by-step, capturing new contextual and structural information after each slide, rather than iterating over fixed feature attention maps (Appendix B, [1]).
>
> **Third, regarding the problem setting and experiments.** Our work focuses on antibody-aware and antibody-agnostic epitope prediction, which is fundamentally different from the TCR–antigen–HLA interaction prediction. Our experimental baselines and evaluation goals are also distinct. We evaluate against multiple cross-attention methods and include ablations to show that sliding attention achieves higher precision in molecular interaction prediction.
>
> Therefore, **we categorically refute the plagiarism allegation** and hope that the reviewer can fairly re-evaluate this concern based on the detailed clarifications we have provided.
>
> ------
>
> [1] Feng, Ziyan, et al. "Sliding-attention transformer neural architecture for predicting T cell receptor–antigen–human leucocyte antigen binding." *Nature Machine Intelligence* 6.10 (2024): 1216-1230.

---

> ### Comment · Reviewer_r9ov · 2025-11-13
> **Thanks for the clarification**
>
> After carefully considering the arguments from the authors, I beleiev that since Feng's paper was cited, this paper should not be categorized as plagiarism, but instead one that applied a 2024 published algorithm to a different application, with the minor revisions of algorithm details. I will revise my ratings shortly. Still, novelty is a major concern for conferences like ICLR.

---

### Official Review · Reviewer_Wy5X · 2025-10-29

**Soundness:** 3
**Presentation:** 3
**Contribution:** 3
**Rating:** 6
**Confidence:** 3

**Summary:**

This paper introduces ABConformer, a sequence-based framework that employs a sliding attention mechanism within a Conformer backbone for accurate antibody–antigen (Ab–Ag) interface prediction without requiring 3D structural data. The model captures both local and global sequence dependencies while incorporating a physics-inspired Gaussian sliding attention that mimics local docking interactions between VH/VL domains and antigens. This design enables fine-grained residue-level contact recovery and improves both paratope and epitope prediction accuracy and efficiency.
Experiments on a recent SARS-CoV-2 Ab–Ag dataset show that ABConformer outperforms existing sequence-based baselines and generalizes well to both antibody-specific and antibody-agnostic prediction tasks.

**Strengths:**

1.	Clear motivation and novelty
The paper addresses an important and challenging problem — antibody–antigen interface prediction from sequence alone — with a clear motivation and innovative approach. The sliding attention mechanism provides both computational efficiency and improved predictive power.
2.	Strong empirical performance
ABConformer achieves superior results over multiple state-of-the-art baselines on widely used benchmark datasets, demonstrating its robustness and practical relevance.
3.	Comprehensive ablation and interpretability
The ablation studies are thorough and convincing. Attention maps align well with known structural interface regions, offering biologically meaningful interpretability. The model also exhibits reasonable generalization in antibody-agnostic scenarios.
4.	High-quality presentation
The manuscript is clearly written, well-structured, and easy to follow. Figures and tables effectively communicate the results.

**Weaknesses:**

1. Experimental scope and robustness
The experiments are generally solid and well-structured, with clear ablation analyses and sensitivity tests that support the model’s main claims. However, although the baseline coverage is fairly comprehensive, several recent and competitive approaches using protein language models such as ESM-IF1, ProtT5, or AlphaBind are not included. Adding these stronger and more contemporary baselines would provide a fairer and more convincing comparison with current SOTA methods. Reporting computational efficiency or variance across folds would further improve the robustness and credibility of the experimental evidence.

2. Generalization across datasets
While the authors include an external evaluation on the SARS-CoV-2, its limited scale and close domain similarity to AACDB reduce the strength of the generalization argument. Additional experiments on heterogeneous datasets, such as HIV or influenza antibody antigen complexes, which would significantly enhance the credibility of the claimed generalization capability.

3. Fusion strategy justification
The separate VH and VL branches followed by fusion are biologically motivated, as VH and VL contribute differentially to paratope formation and contact distinct regions of the antigen surface. Nevertheless, the rationale for choosing this particular fusion strategy should be explicitly stated and compared with possible alternatives to enhance clarity and justification.

**Questions:**

NA

---

> ### Author Response · Authors · 2025-11-28
> **Response to Reviewer Wy5X**
>
> Dear Reviewer Wy5X,
>
> We sincerely thank you for noting the strengths of our work. Following your suggestions, we have  added comparison experiments on the HIV test set and included a new structural prediction method, AlphaFold3, followed by docking into our analyses. All corresponding modifications have been incorporated into the main text.

---

> ### Author Response · Authors · 2025-11-28
> **Response to Reviewer Wy5X for Weaknesses [1/2]**
>
> **Response to W1**
> > Recent and competitive approaches using protein language models such as ESM-IF1, ProtT5, or AlphaBind are not included.
>
> We appreciate the reviewer’s suggestion to include baselines such as ESM-IF1 [1], ProtT5 [2] and AlphaBind [3].
>
> We agree that incorporating ESM-IF1 into antibody–antigen interface prediction is an interesting direction. However, ESM-IF1 does not natively perform interface prediction; it requires backbone coordinates and additional steps to convert sequence-recovery likelihoods into residue-level interface scores. Existing methods such as SEMA-3D [4] and DiscoTope-3.0 [5] use ESM-IF1 specifically for antigen-agnostic epitope prediction. We therefore view ESM-IF1 as valuable future work. For ProtT5, it could in principle be used in the same way as ESM-2, so we consider it another way of embedding rather than a direct baseline for interface prediction. AlphaBind learns quantitative affinities for parental antibodies and is designed for affinity optimization; it does not provide residue-wise prediction, which our benchmarks require.
>
> **Response to W3**
>
> > The rationale for choosing this particular fusion strategy should be explicitly stated and compared with possible alternatives to enhance clarity and justification.
>
> We thank the reviewer for this comment. Our fusion strategy allows the antigen’s attention to be modulated by both heavy and light chains, implemented by linear weighting of the Ag<–>H and Ag<–>L attention maps. In *Appendix. Sensitivity Analysis*, we varied the H/L weights and observed that slightly favoring the heavy chain improves epitope recall. We agree that exploring alternative fusion strategies is a valuable direction for future work.
>
> ---
>
> **References**
>
> [1] Hsu, Chloe, et al. "Learning inverse folding from millions of predicted structures." *International conference on machine learning*. PMLR, 2022.
>
> [2] Elnaggar, Ahmed, et al. "Prottrans: Toward understanding the language of life through self-supervised learning." *IEEE transactions on pattern analysis and machine intelligence* 44.10 (2021): 7112-7127.
>
> [3] Agarwal, Aditya A., et al. "AlphaBind, a domain-specific model to predict and optimize antibody–antigen binding affinity." *Mabs*. Vol. 17. No. 1. Taylor & Francis, 2025.
>
> [4] Shashkova, Tatiana I., et al. "SEMA: Antigen B-cell conformational epitope prediction using deep transfer learning." *Frontiers in immunology* 13 (2022): 960985.
>
> [5] Høie, Magnus Haraldson, et al. "DiscoTope-3.0: improved B-cell epitope prediction using inverse folding latent representations." *Frontiers in immunology* 15 (2024): 1322712.

---

> > ### Author Response · Authors · 2025-11-28
> > **Response to Reviewer Wy5X for Weaknesses [2/2]**
> >
> > **Response to W2**
> > >  Additional experiments on heterogeneous datasets, such as HIV or influenza antibody-antigen complexes, would significantly enhance the credibility of the claimed generalization capability.
> >
> > We have further examined the antibody-antigen interface prediction on the curated HIV dataset obtained from CATNAP [1] at LANL [2]. The results are shown below and included in *Appendix*. Results show that ABConformer achieves higher precision, MCC, PCC and PR-AUC, while maintaining competitive recall on paratope prediction, indicating strong overall performance and broad generalizability.
> >
> > Method         | (Epi) | Prec   | Rec   | MCC   | PCC   | PR-AUC | (Para) | Prec   | Rec   | MCC   | PCC   | PR-AUC
> > ----------------|------|-----|-------|-------|-------|-----------|------|--------|-------|-------|-------|-----------
> > PECAN          |  | 0.579  | 0.635 | 0.578 | 0.575 | 0.590 | | 0.353  | 0.537 | 0.411 | 0.439 | 0.383
> > Honda           | | 0.603  | 0.667 | 0.597 | 0.598 | 0.621 | | 0.333  | 0.570 | 0.405 | 0.436 | 0.378
> > Epi-EPMP      |   | 0.597  | 0.650 | 0.590 | 0.585 | 0.611 | | 0.319  | 0.559 | 0.395 | 0.426 | 0.367
> > PeSTo           | | 0.589  | 0.618 | 0.574 | 0.569 | 0.581 | | 0.325  | 0.491 | 0.385 | 0.416 | 0.359
> > MIPE            | | 0.613  | **0.675** | 0.609 | 0.603 | 0.627 | | 0.401  | 0.563 | 0.460 | 0.487 | 0.413
> > DeepInterAware |  | 0.605  | 0.648 | 0.594 | 0.591 | 0.619 | | 0.339  | 0.565 | 0.408 | 0.441 | 0.383
> > Epi4Ab         |  | -      | -     | -     | -     | -    |  | 0.358  | 0.550 | 0.415 | 0.449 | 0.397
> > AF2 Multimer   |  | 0.483  | 0.639 | 0.525 | -     | -    |  | 0.277  | 0.449 | 0.350 | -     | -
> > AF3            |  | 0.539  | 0.663 | 0.568 | -     | -    |  | 0.313  | 0.534 | 0.391 | -     | -
> > ABConformer    |  | **0.638**  | 0.657 | **0.613** | **0.615** | **0.634** | | **0.407** | **0.585** | **0.473** | **0.498** | **0.424**
> >
> > -----
> >
> > **References**
> >
> > [1] Yoon, Hyejin, et al. "CATNAP: a tool to compile, analyze and tally neutralizing antibody panels." *Nucleic acids research* 43.W1 (2015): W213-W219.
> >
> > [2] Foley, Brian Thomas, et al. *HIV sequence compendium 2018*. No. LA-UR-18-25673. Los Alamos National Laboratory (LANL), Los Alamos, NM (United States), 2018.

---

### Official Review · Reviewer_KNTa · 2025-10-31

**Soundness:** 2
**Presentation:** 2
**Contribution:** 2
**Rating:** 4
**Confidence:** 4

**Summary:**

This paper presents ABConformer, a sequence-based deep learning model for predicting antibody-antigen binding interfaces. The method builds upon the Conformer architecture and introduces a sliding attention mechanism adapted from prior work to simulate molecular docking processes. The model processes antibody heavy chain, light chain, and antigen sequences separately using ESM-2 embeddings as input features. The sliding attention mechanism iteratively adjusts relative positions between sequences through Gaussian kernels for spatial proximity and attention-weighted position updates. The authors evaluate their method on AACDB and an external SARS-CoV-2 dataset, demonstrating improvements over existing sequence-based methods in antibody-specific interface prediction and competitive performance for antibody-agnostic epitope prediction, though with notable limitations in recall and methodological novelty.

**Strengths:**

**S1**. While the sliding attention mechanism is not original to this work, the authors make appropriate adaptations to the antibody-antigen context through a three-branch architecture. The dual sliding process and linear combination of resulting embeddings represent sensible design choices for this specific application. However, the novelty lies primarily in the application domain rather than in methodological innovation.

**S2**. The explicit separation of Ab-H and Ab-L chains is more biologically grounded than treating antibodies as monolithic entities, as paratopes are formed by complementarity-determining regions (CDRs) from both heavy and light chains. This design choice aligns well with the known structural organization of antibody-antigen interfaces.

**S3**. The paper provides thorough experimental evaluation, including extensive ablation studies examining encoding strategies and attention mechanisms, sensitivity analyses of hyperparameters, and case studies.

**Weaknesses:**

**W1**.The paper does not adequately justify how binding site prediction translates to practical antibody design applications. Specifically: (i) How does interface prediction assist in binding affinity estimation, which is often the ultimate goal in therapeutic antibody development? (ii) What is the relationship between predicted interface residues and functional properties? (iii) In what scenarios would researchers prefer interface prediction over structure prediction followed by docking simulations? Without addressing these questions, the practical motivation for this work remains insufficiently justified.

**W2**. The paper proceeds directly to methodology without establishing a rigorous mathematical formulation of the prediction task. A complete problem statement should specify: (i) input, (ii) output, and (iii) objective.

**W3**. The sliding attention mechanism is directly adapted from previous work, with the primary modifications being: (i) extension from two-component to three-branch architecture, (ii) sequential dual sliding operations (Ag→Ab-H, then Ag→Ab-L), and (iii) linear combination of embeddings after each sliding step. The paper should more explicitly distinguish between novel contributions and adaptations of existing methods. Given the limited algorithmic innovation, the contribution might be more accurately characterized as application engineering rather than methodological advancement.

**W4**. The assertion that ABConformer *surpass widely used sequence-based methods* is misleading. According to Table 1 and Figure 9D, ABConformer achieves superior performance only on certain metrics while showing substantially lower recall. In addition, the paper acknowledges that *the sliding-attention module has no effect* for antibody-agnostic prediction because antibody embeddings are set to zero, which reduces ABConformer to a Conformer backbone. This raises the question: why not develop and compare a dedicated Conformer-only model for antibody-agnostic prediction?

**W5**. While the paper includes AlphaFold2 Multimer v3 as a baseline, it omits several highly relevant and state-of-the-art structure-based approaches: (i) AlphaFold3 [1] has demonstrated strong performance on protein-related tasks; (ii) ESMFold [2] uses the same ESM-2 embeddings as ABConformer's encoder, making it a critical baseline for isolating the value added by the sliding attention mechanism; (iii) Boltz-1 [3] and Boltz-2 [4] are recent methods specifically designed for biomolecular interaction modeling and binding affinity prediction; (iv) PAbFold [5] is specifically designed for antibody epitope prediction using AF2. Without these comparisons, the paper cannot substantiate its implicit claim that sequence-based prediction with sliding attention offers advantages over structure prediction followed by interface extraction.

---

**Reference*

[1] J. Abramson et al. *Accurate structure prediction of biomolecular interactions with AlphaFold 3*. Nature 2024.

[2] Z. Lin et al. *Evolutionary-scale prediction of atomic-level protein structure with a language model*. Science 2023.

[3] J. Wohlwend et al. *Boltz-1 democratizing biomolecular interaction modeling*. BioRxiv 2025.

[4] S. Passaro et al. *Boltz-2: Towards accurate and efficient binding affinity prediction*. BioRxiv 2025.

[5] J. DeRoo et al. *PAbFold: linear antibody epitope prediction using AlphaFold2*. BioRxiv 2024.

**Questions:**

**Q1**. How does binding site prediction assist in practical antibody design workflows? Specifically: (i) Can predicted interfaces be used to estimate binding affinity, and if so, what is the correlation? (ii) How might researchers use these predictions in therapeutic antibody optimization or vaccine design?

**Q2**. How does ABConformer compare to a baseline that computes pairwise cosine or Euclidean distances between ESM-2 embeddings of antibody and antigen residues, followed by thresholding? This comparison would isolate the value added by the sliding attention mechanism beyond embedding similarity.

**Q3**. Why are results of Epi4Ab omitted in Table 1?

**Details Of Ethics Concerns:**

No concerns

---

> ### Author Response · Authors · 2025-11-28
> **Response to Reviewer KNTa**
>
> Dear Reviewer KNTa,
>
> We sincerely appreciate the time and effort you have dedicated to reviewing our manuscript and providing valuable suggestions. After carefully reading your comments, we have included the mathematical description of our task, rebuilt a pure Conformer in the absence of sliding attention, added comparison experiments on the HIV test set, and included a new structural prediction method, AlphaFold3, followed by docking into our analyses. All corresponding modifications have been incorporated into the main text. We also responded to all proposed weaknesses and questions.

---

> ### Author Response · Authors · 2025-11-28
> **Response to Reviewer KNTa for Weaknesses [1/3]**
>
> **Response to W1**
>
> > W1: (i) How does interface prediction assist in binding affinity estimation, which is often the ultimate goal in therapeutic antibody development?
>
> The primary goal in therapeutic antibody development is often to improve binding affinity. In practice, directly evaluating affinity for all possible mutations is computationally expensive and experimentally infeasible, because the combinatorial space of CDR mutations is very large. Interface prediction helps by identifying which residues are most likely to participate in antigen binding. By focusing subsequent mutational design and structural refinement on these residues, the search space is greatly reduced.
>
> Recent sequence-based model, ProtAttnBA [1], shows that even approximate interface predictions can help guide affinity estimation by focusing subsequent modeling on the most relevant residues. Deep mutational scanning studies of antibody–antigen complexes also indicate that many mutations affecting binding or escape are located at or near interface residues, with a few functionally important mutations occur outside the interface [2,3]. Focusing mutagenesis or modeling efforts on predicted interface residues therefore increases the likelihood of identifying changes that significantly impact affinity.
>
> > (ii) What is the relationship between predicted interface residues and functional properties?
>
> Predicted interface residues correspond to amino acids most likely to contact the binding partner. Experimental studies show that mutations at these residues disproportionately affect binding affinity and neutralization potency, as well as escape from antibodies [2,3]. Thus, predicted interfaces provide a residue-level map that highlights positions most critical for functional outcomes, enabling more efficient targeting in design or mutagenesis.
>
> > (iii) In what scenarios would researchers prefer interface prediction over structure prediction followed by docking simulations?
>
> While structure prediction followed by docking is a common approach, current tools often fail to produce accurate antibody–antigen structures. For instance, benchmarks of AlphaFold 3 show that high‑accuracy docking is achieved in only around 10–13% of antibody or nanobody complexes [4]. Broader benchmarking of AlphaFold on diverse protein–protein complexes shows that, although many heterodimeric complexes are predicted near‑native with top-ranked models (43% success), antibody–antigen complexes are poorly modeled, with only about 11% of predictions achieving near‑native accuracy using AlphaFold-Multimer [5]. Moreover, we evaluated interface extraction using AlphaFold2 Multimer and newly performed AlphaFold3 predictions. In both cases, the overall performance was substantially lower than that achieved by state-of-the-art deep learning models designed for antibody–antigen interface prediction (see updated *Comparison Experiments*).
>
> **Response to W2**
> > W2: The paper proceeds directly to methodology without establishing a rigorous mathematical formulation of the prediction task that specify (i) input, (ii) output, and (iii) objective.
>
> We appreciate the reviewer’s suggestion. In our task, both paratope and epitope prediction are formulated as sequence-based residue-level classification problems. The antibody-specific mode takes (𝐻, 𝐿, 𝐴) as input, and the antibody-agnostic mode only uses (𝐴). The model outputs a probability in [0,1] for each residue, representing its predicted interface likelihood. The learning objective is the standard residue-wise binary cross-entropy, which we have illustrated in *Appendix-Training Details* before. Since the mathematical formulation is straightforward, we have added a *Problem Statement* section in the appendix of the revised manuscript to further clarify inputs, outputs and objectives.
>
> -----
> **References**
>
> [1] Liu, Chen, et al. "Sequence-only prediction of binding affinity changes: a robust and interpretable model for antibody engineering." *Bioinformatics* 41.8 (2025): btaf446.
>
> [2] Starr, Tyler N., et al. "Deep mutational scanning of SARS-CoV-2 receptor binding domain reveals constraints on folding and ACE2 binding." *cell* 182.5 (2020): 1295-1310.
>
> [3] Greaney, Allison J., et al. "Mapping mutations to the SARS-CoV-2 RBD that escape binding by different classes of antibodies." *Nature communications* 12.1 (2021): 4196.
>
> [4] Hitawala, Fatima N., and Jeffrey J. Gray. "What does AlphaFold3 learn about antibody and nanobody docking, and what remains unsolved?." *MAbs*. Vol. 17. No. 1. Taylor & Francis, 2025.
>
> [5] Yin, Rui, et al. "Benchmarking AlphaFold for protein complex modeling reveals accuracy determinants." *Protein Science* 31.8 (2022): e4379.

---

> ### Author Response · Authors · 2025-11-28
> **Response to Reviewer KNTa for Weaknesses [2/3]**
>
> **Response to W3**
> > W3: The paper should more explicitly distinguish between novel contributions and adaptations of existing methods.
>
> While ABConformer builds upon previous sliding attention mechanisms [1], several key modifications were introduced to address the general context of antibody–antigen complexes. We implement dynamic bandwidths (Eq.3) and normalization strategies to handle variable sequence lengths, and update feature embeddings step-by-step during the sliding process, capturing new contextual and structural information after each slide rather than iterating over fixed attention maps (Appendix B).
>
> Importantly, most prior work on modeling molecular interactions relies on cross attention. Our experiments demonstrate that the sliding attention mechanism, by integrating both spatial attention and position updates, achieves considerably higher modelling accuracy, a result not previously reported.
>
> -------
> [1] Feng, Ziyan, et al. "Sliding-attention transformer neural architecture for predicting T cell receptor–antigen–human leucocyte antigen binding." *Nature Machine Intelligence* 6.10 (2024): 1216-1230.
>
> -----
> **Response to W4**
> >W4: (i) The assertion that ABConformer *surpass widely used sequence-based methods* is misleading.
>
> We appreciate the reviewer for this comment. We have revised the text to avoid overstating the results. ABConformer is designed for antibody-specific interface prediction, and its geometry-based updates favor residue pairs that are spatially proximate with antibody–antigen interactions. This leads to higher precision and corresponding gains in global metrics such as MCC, PCC and PR-AUC. The lower recall reflects that our model avoids assigning contacts outside feasible geometric ranges, and in practice, this produces more reliable interface prediction. Furthermore, we have rebuilt a pure Conformer for antibody-agnostic epitope prediction, which more convincingly demonstrates this assertion. Please see the new results in the next section.
>
> > (ii)  why not develop and compare a dedicated Conformer-only model for antibody-agnostic prediction?
>
> We appreciate the reviewer for this suggestion. For the antibody-agnostic epitope prediction, our aim was not to promote ABConformer as a general sequence-only predictor. Antibody information is removed to disable the sliding-attention module and allows us to isolate its contribution to antibody-conditioned interaction learning. New results are shown in table 1.
>
> To make the antibody-agnostic task clearer, we rebuilt a pure Conformer (AGConformer) trained on antigen information and compared it with widely used methods, as shown below. Results show that the pure Conformer built on ESM-2 embeddings outperforms commonly used sequence-based epitope predictors. We have restated this part and demonstrated these results in *Comparison Experiments*.
>
> -----
> | Method          |   F1   |   MCC   |   PCC   |  PR-AUC |
> |-----------------|--------|---------|---------|---------|
> | BepiPred-3.0    | 0.143  |  0.162  |  0.187  |  0.094  |
> | SEMA-1D 2.0     | 0.152  |  0.164  |  0.195  |  0.107  |
> | DiscoTope-3.0   | 0.277  |  0.273  |  0.325  |  0.231  |
> | AGConformer     | **0.311** | **0.298** | **0.336** | **0.245** |
>
>
> Table 1. Benchmark comparison of antibody-agnostic epitope prediction on the SARS-CoV-2 test set.

---

> ### Author Response · Authors · 2025-11-28
> **Response to Reviewer KNTa for Weaknesses [3/3]**
>
> **Response to W5**
> >W5: This work omits several highly relevant and state-of-the-art structure-based approaches.
>
> We appreciate the reviewer for highlighting additional baselines. We have added experiments using AlphaFold3 (AF3) [1]. However, we note that not all suggested baselines are suitable and necessary for comparison.
>
> First, ESMFold was originally developed as a protein language model for monomeric structure prediction [2]. Previous studies applying ESMFold to protein–peptide docking have shown that its overall performance is generally lower than that of AF2 Multimer and AF3 [3]. Therefore, ESMFold is not suitable for antibody–antigen interface prediction. We have instead focused on AF2 Multimer and AF3, which provide sufficient structure-based benchmarks.
>
> Second, Boltz‑1 [4] and Boltz‑2 [5] are energy- and scoring-based models developed for global binding affinity estimation. They do not generate residue-level interface probabilities and cannot be used for either antibody-specific interface prediction or antibody-agnostic epitope prediction. No peer-reviewed work demonstrates that these models can produce per-residue interface predictions. As such, they are not appropriate baselines for our study.
>
> Third, PAbFold is specifically designed for linear epitope prediction, modeling short peptide–scFv complexes and evaluating binding likelihood based on per-peptide confidence scores [6]. However, our work targets general Ab-Ag interface prediction, including both linear and conformational epitopes [7] and corresponding paratopes. Because linear and conformational epitope prediction address different aspects of antigen recognition, a direct comparison would not be fair.
>
>
> **Here we present the results from AF3 on SARS-CoV-2 test set.** Across both paratope and epitope tasks, ABConformer achieves higher precision and MCC of interface predcition than both structure prediction methods.
>
> ------
> |   (paratope)    |   Prec   |   Rec   |   MCC   |
> |-----------------|----------|---------|---------|
> |   AF2 Multimer  |   0.527  |  0.630  |  0.542  |
> |       AF3       |   0.576  |  **0.649**  |  0.593  |
> |   ABConformer   |   **0.693**  |  0.613  |  **0.622**  |
>
> |    (epitope)    |   Prec   |   Rec   |   MCC   |
> |-----------------|----------|---------|---------|
> |   AF2 Multimer  |   0.275  |  0.496  |  0.307  |
> |       AF3       |   0.315  |  0.518  |  0.383  |
> |   ABConformer   |   **0.467**  |  **0.545**  |  **0.492**  |
>
> Table 2. Comparison with structure-based methods on the SARS-CoV-2 test set
>
> ------
> **References**
>
> [1] Abramson, Josh, et al. "Accurate structure prediction of biomolecular interactions with AlphaFold 3." *Nature* 630.8016 (2024): 493-500.
>
> [2] Lin, Zeming, et al. "Evolutionary-scale prediction of atomic-level protein structure with a language model." *Science* 379.6637 (2023): 1123-1130.
>
> [3] Zalewski, Mateusz, Bjorn Wallner, and Sebastian Kmiecik. "Protein–peptide docking with ESMFold language model." *Journal of Chemical Theory and Computation* 21.6 (2025): 2817-2821.
>
> [4] Wohlwend, Jeremy, et al. "Boltz-1 democratizing biomolecular interaction modeling." *BioRxiv* (2025): 2024-11.
>
> [5] Passaro, Saro, et al. "Boltz-2: Towards accurate and efficient binding affinity prediction." *BioRxiv* (2025).
>
> [6] DeRoo, Jacob, et al. "PAbFold: linear antibody epitope prediction using AlphaFold2." *bioRxiv* (2024).
>
> [7] Forsström, Björn, et al. "Dissecting antibodies with regards to linear and conformational epitopes." *PloS one* 10.3 (2015): e0121673.

---

> > ### Author Response · Authors · 2025-11-28
> > **Response to Reviewer KNTa for Questions [1/1]**
> >
> > **Response to Q1**
> > >(i) Can predicted interfaces be used to estimate binding affinity, and if so, what is the correlation? (ii) How might researchers use these predictions in therapeutic antibody optimization or vaccine design?
> >
> > We appreciate the reviewer’s follow-up question. As noted in *response to Weakness 1*, predicted interfaces help identify residues that are most likely to contact the antigen, which allows researchers to focus mutagenesis or affinity-tuning efforts on a smaller and more relevant part of the sequence. Experimental studies such as deep mutational scanning show that changes at interface residues have a disproportionate effect on binding and neutralization [1,2]. Therefore, the predicted interfaces are highly informative for affinity optimization. We have clarified in the main text how our task supports downstream antibody optimization.
> >
> > **Response to Q2**
> > > How does ABConformer compare to a baseline that computes pairwise cosine or Euclidean distances between ESM-2 embeddings of antibody and antigen residues, followed by thresholding?
> >
> > We understand the reviewer’s motivation for suggesting this baseline. However, this approach poorly capture antibody–antigen interactions because ESM-2 encodes each chain independently. The embeddings contain only single-chain contextual information; they do not describe whether two residues can come into contact, whether the antigen residue is compatible with the specific antibody, or how one chain influences the other. When cosine or Euclidean distances are computed, the resulting matrix mostly reflects generic similarity patterns that leads to a high false-positive rate.
> >
> > This is why, to our knowledge, raw ESM‑2 distance matrices have not been used as a single baseline for extended applications. Instead, recent antibody–antigen predictors using ESM‑2 embeddings integrate explicit mechanisms to model contacts between antibodies and antigens. For instance, MambaAAI [3] employs a dual-view Mamba encoder, and EPP [4] incorporates a Bi-LSTM.
> >
> > **Response to Q3**
> > >Why are results of Epi4Ab omitted in Table 1?
> >
> > Epi4Ab [5] is designed to identify antigen-side epitope residues conditioned on antibody VH/VL families and CDR sequences; it does not include paratope prediction.
> >
> > ----
> > **References**
> >
> > [1] Starr, Tyler N., et al. "Deep mutational scanning of SARS-CoV-2 receptor binding domain reveals constraints on folding and ACE2 binding." *cell* 182.5 (2020): 1295-1310.
> >
> > [2] Greaney, Allison J., et al. "Mapping mutations to the SARS-CoV-2 RBD that escape binding by different classes of antibodies." *Nature communications* 12.1 (2021): 4196.
> >
> > [3] Liu, Xuan, et al. "Bio-Inspired Mamba for Antibody–Antigen Interaction Prediction." *Biomolecules* 15.6 (2025): 764.
> >
> > [4] Li, Qianying, et al. "Enhanced prediction of antigen and antibody binding interface using ESM-2 and Bi-LSTM." *Human Immunology* 86.3 (2025): 111304.
> >
> > [5] Tran, Nhan Dinh, Krithika Subramani, and Chinh Tran-To Su. "Epi4Ab: a data-driven prediction model of conformational epitopes for specific antibody VH/VL families and CDRs sequences." *mAbs*. Vol. 17. No. 1. Taylor & Francis, 2025.

---

### Official Review · Reviewer_qZkx · 2025-11-01

**Soundness:** 3
**Presentation:** 3
**Contribution:** 2
**Rating:** 4
**Confidence:** 3

**Summary:**

This paper presents ABConformer, a new model that predicts antibody-antigen binding interfaces using only their protein sequences. The  key innovation is a physics-inspired sliding attention mechanism, which aims to mimic the molecular docking process by sliding the antigen sequence against the antibody's heavy and light chains to find stable interaction patterns, without the need for 3D structural data.

**Strengths:**

- The physics-inspired sliding attention is an interesting innovation, which outperforms conventional cross-attention, especially in improving the precision of epitope prediction.
- The model is flexible, and can handle antibody-specific prediction (when all sequences are known) while also being effective for antibody-agnostic pan-epitope prediction (using only the antigen sequence), where it surpasses other sequence-based methods.

**Weaknesses:**

- The core innovation of this paper is the sliding attention module, which isn't used for the antibody-agnostic predictions.
- Comparisons are performed on a small external dataset of 35 SARS-CoV-2 complexes, which raises questions about generalizability. ABconformer is outperformed by DiscoTope-3.0, a structure-based method on the antibody-agnostic task. Recent structure-based methods such as AlphaFold3, Boltz-2 or Chai are also not included in the antibody-specific comparisons.
- The ablation study shows that the gap between sliding attention and conventional cross-attention is small, with cross-attention achieving slightly higher recall, but worse precision.

**Questions:**

- Could the authors include other benchmarks than SARS-CoV-2?
- It would be valuable to include a comparison with AlphaFold3, or other recent structure prediction models.

---

> ### Author Response · Authors · 2025-11-28
> **Response to Reviewer qZkx**
>
> Dear Reviewer qZkx,
>
> We appreciate all of your valuable and encouraging comments. After carefully reading your suggestions, we have rebuilt a pure Conformer in the absence of sliding attention, added comparison experiments on the HIV test set and included a new structural prediction method, AlphaFold3, followed by docking into our analyses. All corresponding modifications have been incorporated into the main text.

---

> > ### Author Response · Authors · 2025-11-28
> > **Response to Reviewer qZkx for Weaknesses [1/2]**
> >
> > **Response to W1**
> > > The core innovation of this paper is the sliding attention module, which isn't used for the antibody-agnostic predictions.
> >
> > Thank you for raising this point. ABConformer is built for antibody-specific interface prediction, and the ablation is only intended to test the effect of removing antibody information and transfer to another biological task (i.e., antigen-only epitope prediction), rather than to position the model as a general sequence-only predictor. Removing antibody information disables the sliding-attention module, allowing us to isolate its contribution to antibody-conditioned interaction learning.
> >
> > To make this point clearer, we rebuilt a pure Conformer (AGConformer) for the antigen-only task and evaluated on the SARS-CoV-2 test set, showing that the pure Conformer built on ESM-2 embeddings outperforms commonly used epitope predictors. We have revised these results in *Comparison Experiments* and reclaimed this task.
> >
> >
> > | Method          |   F1   |   MCC   |   PCC   |  PR-AUC |
> > |-----------------|--------|---------|---------|---------|
> > | BepiPred-3.0    | 0.143  |  0.162  |  0.187  |  0.094  |
> > | SEMA-1D 2.0     | 0.152  |  0.164  |  0.195  |  0.107  |
> > | DiscoTope-3.0   | 0.277  |  0.273  |  0.325  |  0.231  |
> > | AGConformer     | **0.311** |  **0.298**  |  **0.336**  |  **0.245**  |

---

> > > ### Author Response · Authors · 2025-11-28
> > > **Response to Reviewer qZkx for Weaknesses [2/2]**
> > >
> > > **Response to W2**
> > >
> > > > W2: (i) Generalizability
> > >
> > > We appreciate the reviewer’s suggestion for more test cases, and we have further examined the antibody-antigen interface prediction on the curated HIV dataset obtained from CATNAP [1] at LANL [2]. The results are shown in *Respond to Q1* and included in *Appendix*.
> > >
> > > > (ii) Other baselines
> > >
> > > We appreciate the reviewer’s suggestion for more recent baselines, and we have further added Alphafold3 into experiments. The results are shown in *Respond to Q2* and added to the *Comparison Experiments*. While Boltz-2 [3] is designed for global binding affinity estimation. They do not generate residue-level interface probabilities and cannot be used for either antibody-specific interface prediction or antibody-agnostic epitope prediction.
> > >
> > > **Response to W3**
> > > > Cross-attention achieves slightly higher recall, but worse precision.
> > >
> > > ABConformer is designed for antibody-specific interface prediction, and its geometry-based updates favor residue pairs that are spatially proximate with antibody–antigen interactions. This leads to higher precision and corresponding gains in metrics such as MCC, PCC and PR-AUC. The lower recall reflects that our model avoids assigning contacts outside feasible geometric ranges, and in practice, this produces more reliable interface prediction. Furthermore, the recall can be adjusted by changing the threshold of extraction.
> > >
> > > ----
> > >
> > > **References**
> > >
> > > [1] Yoon, Hyejin, et al. "CATNAP: a tool to compile, analyze and tally neutralizing antibody panels." *Nucleic acids research* 43.W1 (2015): W213-W219.
> > >
> > > [2] Foley, Brian Thomas, et al. *HIV sequence compendium 2018*. No. LA-UR-18-25673. Los Alamos National Laboratory (LANL), Los Alamos, NM (United States), 2018.
> > >
> > > [3] Passaro, Saro, et al. "Boltz-2: Towards accurate and efficient binding affinity prediction." *BioRxiv* (2025).

---

> > > > ### Author Response · Authors · 2025-11-28
> > > > **Response to Reviewer qZkx for Questions [1/2]**
> > > >
> > > > **Response to Q1**
> > > > >Could the authors include other benchmarks than SARS-CoV-2?
> > > >
> > > > Yes! We have further examined the antibody-antigen interface prediction on the curated HIV dataset. The results are shown below and included in *Appendix*. Results show that ABConformer attains higher precision, MCC, PCC and PR-AUC, while maintaining competitive recall on paratope prediction, indicating strong overall performance and broad generalizability.
> > > >
> > > >
> > > >
> > > > Method         | (Epi) | Prec   | Rec   | MCC   | PCC   | PR-AUC | (Para) | Prec   | Rec   | MCC   | PCC   | PR-AUC
> > > > ----------------|------|-----|-------|-------|-------|-----------|------|--------|-------|-------|-------|-----------
> > > > PECAN          |  | 0.579  | 0.635 | 0.578 | 0.575 | 0.590 | | 0.353  | 0.537 | 0.411 | 0.439 | 0.383
> > > > Honda           | | 0.603  | 0.667 | 0.597 | 0.598 | 0.621 | | 0.333  | 0.570 | 0.405 | 0.436 | 0.378
> > > > Epi-EPMP      |   | 0.597  | 0.650 | 0.590 | 0.585 | 0.611 | | 0.319  | 0.559 | 0.395 | 0.426 | 0.367
> > > > PeSTo           | | 0.589  | 0.618 | 0.574 | 0.569 | 0.581 | | 0.325  | 0.491 | 0.385 | 0.416 | 0.359
> > > > MIPE            | | 0.613  | **0.675** | 0.609 | 0.603 | 0.627 | | 0.401  | 0.563 | 0.460 | 0.487 | 0.413
> > > > DeepInterAware |  | 0.605  | 0.648 | 0.594 | 0.591 | 0.619 | | 0.339  | 0.565 | 0.408 | 0.441 | 0.383
> > > > Epi4Ab         |  | -      | -     | -     | -     | -    |  | 0.358  | 0.550 | 0.415 | 0.449 | 0.397
> > > > AF2 Multimer   |  | 0.483  | 0.639 | 0.525 | -     | -    |  | 0.277  | 0.449 | 0.350 | -     | -
> > > > AF3            |  | 0.539  | 0.663 | 0.568 | -     | -    |  | 0.313  | 0.534 | 0.391 | -     | -
> > > > ABConformer    |  | **0.638**  | 0.657 | **0.613** | **0.615** | **0.634** | | **0.407** | **0.585** | **0.473** | **0.498** | **0.424**

---

> > > > > ### Author Response · Authors · 2025-11-28
> > > > > **Response to Reviewer qZkx for Questions [2/2]**
> > > > >
> > > > > **Response to Q2**
> > > > >
> > > > > > It would be valuable to include a comparison with AlphaFold3, or other recent structure prediction models.
> > > > >
> > > > > We thank the reviewer's suggestion for including state-of-the-art structure prediction methods followed by docking. In response, we have included AlphaFold3 as a baseline for both SARS-CoV-2 and HIV test sets. HIV results are newly added and demonstrated in *Response to Q1*. Here, we show the SARS-CoV-2 results, where ABConformer outperforms both structure-based methods in terms of precision and MCC for interface prediction.
> > > > >
> > > > >
> > > > > |   (paratope)    |   Prec   |   Rec   |   MCC   |
> > > > > |-----------------|----------|---------|---------|
> > > > > |   AF2 Multimer  |   0.527  |  0.630  |  0.542  |
> > > > > |       AF3       |   0.576  |  **0.649**  |  0.593  |
> > > > > |   ABConformer   |   **0.693**  |  0.613  |  **0.622**  |
> > > > >
> > > > > |    (epitope)    |   Prec   |   Rec   |   MCC   |
> > > > > |-----------------|----------|---------|---------|
> > > > > |   AF2 Multimer  |   0.275  |  0.496  |  0.307  |
> > > > > |       AF3       |   0.315  |  0.518  |  0.383  |
> > > > > |   ABConformer   |   **0.467**  |  **0.545**  |  **0.492**  |

---

### Meta-Review · Area_Chair_KHDu · 2025-12-04

**Summary:**

This paper presents ABConformer, a new model that predicts antibody-antigen binding interfaces using only their protein sequences. The key innovation is a physics-inspired sliding attention mechanism, which aims to mimic the molecular docking process by sliding the antigen sequence against the antibody's heavy and light chains to find stable interaction patterns.

**Reviewer Concerns:**

Most reviewers think it is below the threshold. One reviewer think it may copy from existing paper, and then he revise his idea after rebuttal. However, the reviewer still think its novelty is not engouh.

**Reviewer Scores:**

I do not think these reviewers will change their scores.

---

### Decision · Program_Chairs · 2026-01-26

Reject